# MIRO: MultI-Reward cOnditioned pretraining improves T2I quality and efficiency

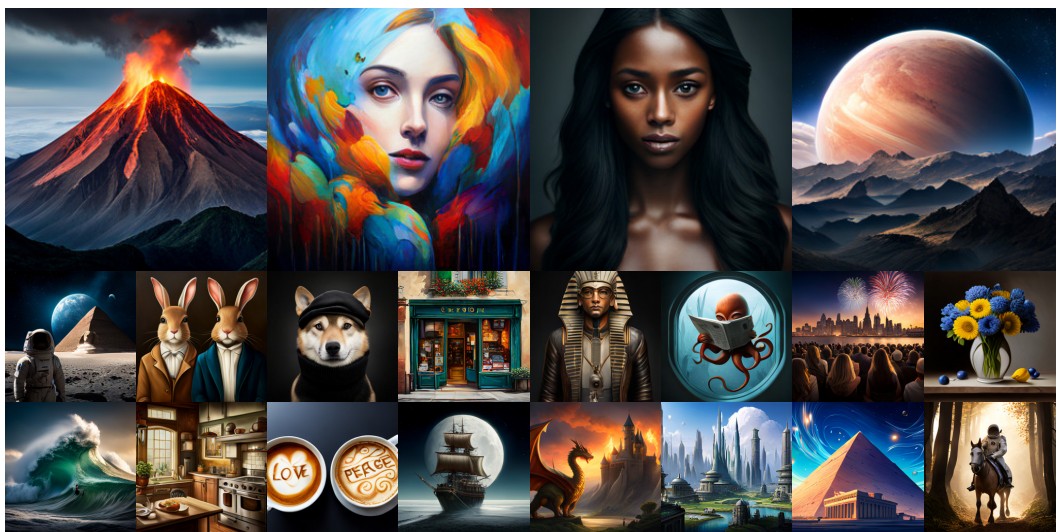

Figure 1: Images from our MIRO Synth model on PartiPrompt(Yu et al., 2022).

## Abstract

Current text-to-image generative models are trained on large uncurated datasets to enable diverse generation capabilities. However, this does not align well with user preferences. Recently, reward models have been specifically designed to perform post-hoc selection of generated images and align them to a reward, typically user preference. This discarding of informative data together with the optimizing for a single reward tend to harm diversity, semantic fidelity and efficiency. Instead of this post-processing, we propose to condition the model on multiple reward models during training to let the model learn user preferences directly. We show that this not only dramatically improves the visual quality of the generated images but it also significantly speeds up the training. Our proposed method, called MIRO, achieves state-of-the-art performances on the GenEval compositional benchmark and user-preference scores (PickAScore, ImageReward, HPSv2).

## 1 Introduction

Aligning with human preferences. How can I apply it to my generative AI problem? There are many success stories in LLMs (Christiano et al., 2017; Rafailov et al., 2023) and even text-to-image generation (Fan et al., 2023), where this alignment has been excelling. In fact, today's best text-to-image generation systems are typically trained in three stages: large-scale pretraining on noisy web data followed by post-hoc alignment using curated subsets and then reinforcement learning from human feedback (RLHF)(Esser et al., 2024; Labs, 2024). While effective, this paradigm carries well-known downsides: it discards informative "low-quality" data (Dufour et al., 2024), complicates training with an additional optimization stage, and tends to overfit to a single reward, often harming diversity (mode collapse) or semantic fidelity and efficiency.

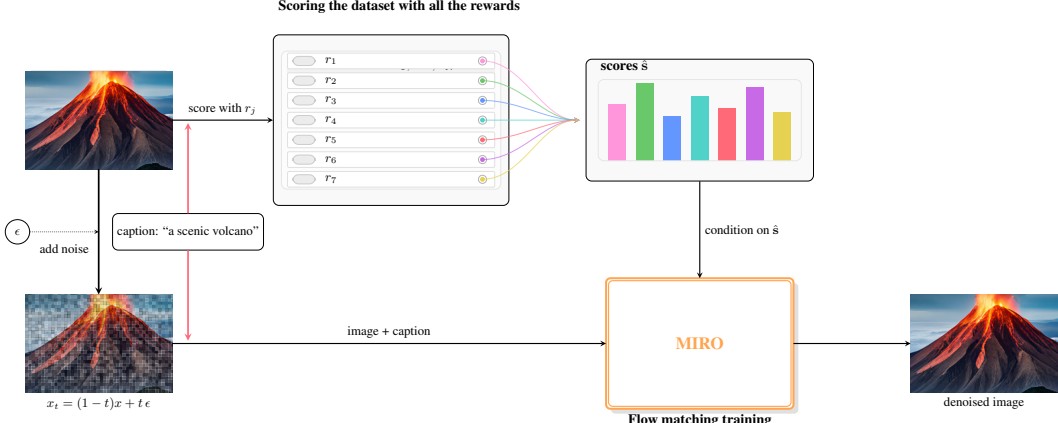

Figure 2: MIRO training pipeline. Top: dataset *scoring* with multiple rewards $r_1, \ldots, r_N$ produces a scores vector $\hat{\mathbf{s}}$. Bottom: during *training*, the model conditions on $\hat{\mathbf{s}}$ and a noisy input $x_t = (1 - t)x + t\,\epsilon$ to learn to denoise toward high-reward regions.

We ask a simple question: "rather than correcting a pre-trained text-to-image model, can we teach it how to trade off multiple rewards *from the beginning*"?

Our answer is MultI-Reward cOnditioning (**MIRO**) pretraining, a framework that integrates multiple reward signals directly into the pretraining objective for text-to-image generation. Similar to Dufour et al. (2024), we condition the generative model on a vector of reward scores per text-image pair, and the model thus learns an explicit mapping from desired reward levels to visual characteristics. The rewards span aesthetics, user preference, semantic correspondence, visual reasoning, and domain-specific correctness.

This simple change has powerful consequences. First, it preserves the full spectrum of data quality instead of filtering it out, allowing the model to learn how different reward levels manifest visually. Second, it turns alignment into a controllable variable at inference time: users can dial individual rewards up or down, or recover a multi-reward analogue of classifier-free guidance that steers towards jointly high-reward regions. Third, by providing rich supervision at scale, **MIRO** accelerates convergence and improves sample efficiency.

Empirically, a small model trained with **MIRO** on a 16M-image setup outperforms no reward conditioning and single-reward baselines: it converges up to $19\times$ faster on AestheticScore (Schuhmann et al., 2022), HPSv2 (Wu et al., 2023), PickScore (Kirstain et al., 2023), and ImageReward (Xu et al., 2023), mitigates reward hacking, and improves compositional alignment. It outperforms much bigger models like Flux-dev on GenEval (Ghosh et al., 2024) and user preference scores (Xu et al., 2023; Wu et al., 2023; Kirstain et al., 2023), while remaining substantially more compute-efficient.

Our contributions are the following:

- We propose **MIRO**: a reward-conditioned pretraining that integrates rewards directly during training alleviating post-hoc processing,
- **MIRO** achieves state-of-the-art scores on GenEval and user-preference metrics, outperforming much bigger models trained for much longer,
- **MIRO** converges up to $19\times$ faster than regular training and achieves the same quality with orders of magnitude less inference compute ($370\times$ less than Flux for example).

## 2  METHOD

We introduce **MultI-Reward cOnditioning Pretraining (MIRO)**, a framework for conditional image generation that incorporates multiple reward signals directly into the pretraining phase. Our key insight is that by conditioning the generative model on explicit reward scores during training, we can preserve the full spectrum of quality levels while enabling fine-grained control over multiple

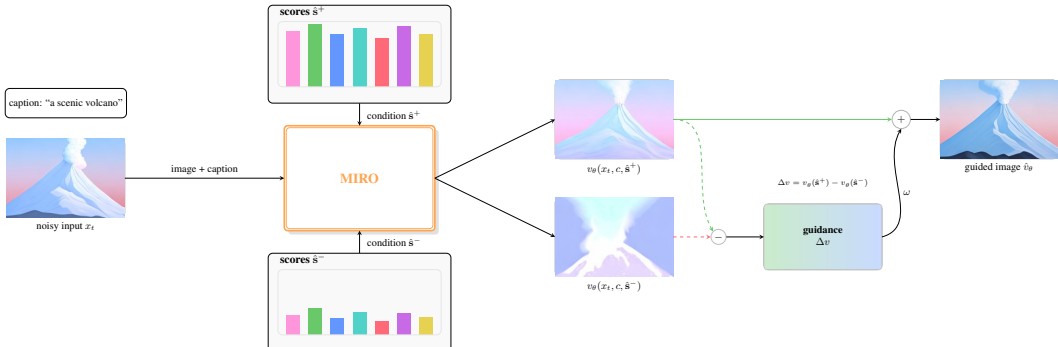

Figure 3: MIRO inference overview (single model). The previous step $x_t$ and caption are fed to one MIRO model while conditioning on two reward histograms: $\hat{\mathbf{s}}^+$ (top) and $\hat{\mathbf{s}}^-$ (bottom), producing $v_\theta(x_t, c, \hat{\mathbf{s}}^+)$ and $v_\theta(x_t, c, \hat{\mathbf{s}}^-)$. The guidance direction $\Delta v = v_\theta(\hat{\mathbf{s}}^+) - v_\theta(\hat{\mathbf{s}}^-)$ is scaled by $\omega$ and added to the high-reward output to obtain the guided image $\hat{v}_\theta$.

objectives at inference time. This approach eliminates the need for separate alignment stages (Fan et al., 2023) while providing unprecedented flexibility in reward trade-offs.

**Method Overview** Our method consists of three key components: (1) **Dataset Augmentation**, where we enrich the pretraining dataset with reward annotations across multiple quality dimensions; (2) **Multi-Reward Conditioned Training**, where we modify the flow matching objective to incorporate reward signals directly into the generative process; and (3) **Reward-Guided Inference** enables fine-grained control over generation quality through explicit reward conditioning during sampling.

**Problem Formulation** Let $\mathcal{D} = \{(x^{(i)}, c^{(i)})\}_{i=1}^M$ be a large-scale pretraining dataset where $x^{(i)} \in \mathbb{R}^{H \times W \times 3}$ represents an image and $c^{(i)} \in \mathcal{T}$ represents the corresponding text condition (e.g., caption, prompt). Traditional pretraining learns a generative model $p_\theta(x|c)$ that captures the joint distribution of images and text without explicit quality control.

In contrast, we consider a set of $N$ reward models $\mathcal{R} = \{r_1, r_2, \ldots, r_N\}$ where each $r_j : \mathbb{R}^{H \times W \times 3} \times \mathcal{T} \to \mathbb{R}$ evaluates different aspects of image quality, with $\mathcal{T}$ being the associated conditioning space.

Our goal is to learn a conditional generative model $p_\theta(x|c, \mathbf{s})$ where $\mathbf{s} = [s_1, s_2, \ldots, s_N]$ represents the desired reward levels, enabling controllable generation across multiple quality dimensions.

### 2.1 DATASET AUGMENTATION WITH REWARD SCORES

The first step of MIRO involves augmenting the pretraining dataset with comprehensive reward annotations. For each sample $(x^{(i)}, c^{(i)}) \in \mathcal{D}$, we compute reward scores across all $N$ reward models:

$$s_j^{(i)} = r_j(x^{(i)}, c^{(i)}) \quad \forall j \in \{1, 2, \ldots, N\} \tag{1}$$

This process transforms our dataset into an enriched version $\tilde{\mathcal{D}} = \{(x^{(i)}, c^{(i)}, \mathbf{s}^{(i)})\}_{i=1}^M$ where $\mathbf{s}^{(i)} = [s_1^{(i)}, s_2^{(i)}, \ldots, s_N^{(i)}]$ contains the multi-dimensional quality assessment for each sample.

**Score Normalization and Binning.** Raw reward scores often exhibit different scales and distributions across reward models, making direct conditioning challenging. To address this, we employ a uniform binning strategy into $B$ bins that ensures balanced representation across quality levels. Details are found in the Supplementary Material.

### 2.2 MULTI-REWARD CONDITIONED FLOW MATCHING

Having augmented our dataset with reward scores, we now incorporate these signals into the generative model architecture. We build upon flow matching Lipman et al. (2023), a powerful framework for training continuous normalizing flows that has shown excellent performance in high-resolution image generation.

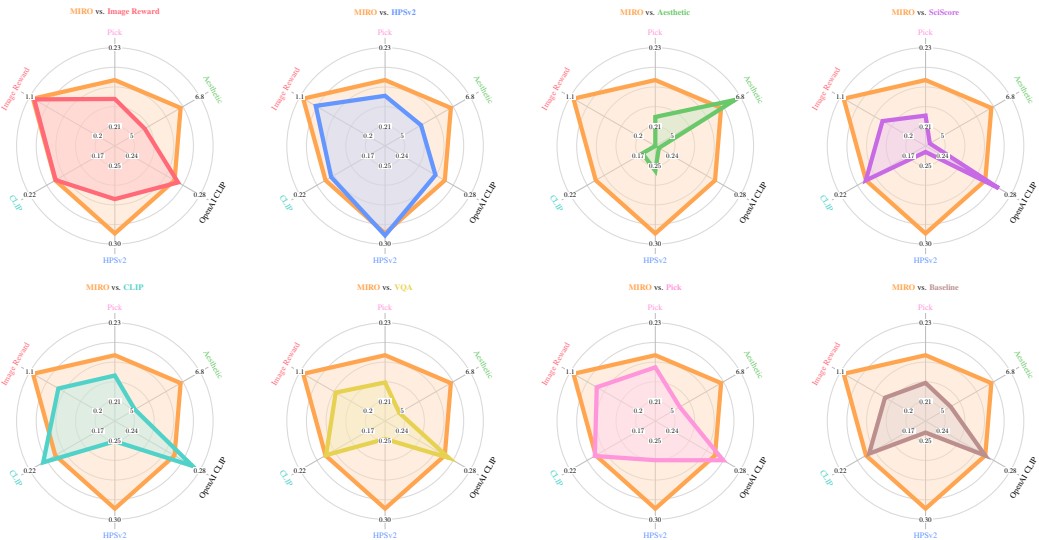

Figure 4: Comparison of the MIRO model against eight other specialist/baseline models. Each radar plot shows **MIRO** versus a comparison model across six metrics.

**Training Objective.** Following the standard flow matching formulation, we sample noise $\epsilon \sim \mathcal{N}(0, I)$ and time $t \sim \mathcal{U}(0, 1)$, then compute the noisy sample $x_t = (1 - t)x + t\epsilon$. The multi-reward flow matching loss becomes:

$$\mathcal{L} = \mathbb{E}_{(x,c,\hat{s})\sim\tilde{\mathcal{D}},\epsilon\sim\mathcal{N}(0,I),t\sim\mathcal{U}(0,1)} \left[ \|v_\theta(x_t, c, \hat{s}) - (\epsilon - x)\|_2^2 \right] \tag{2}$$

This objective trains the model to predict the difference between the noise and the clean image, conditioned on both the text prompt and the desired quality levels. The model learns to associate different reward levels with corresponding visual characteristics, enabling reward-aware generation.

**Training Dynamics.** During training, the model observes the full spectrum of quality levels for each reward dimension. This exposure allows it to learn the relationship between reward values and visual features, from low-quality samples that may exhibit artifacts or poor composition to high-quality samples with superior aesthetics and text alignment.

## 2.3 INFERENCE WITH REWARD-GUIDED SAMPLING

At inference time, MIRO provides unprecedented control over the generation process through explicit reward conditioning. This section details the various sampling strategies enabled by our approach.

**High-Quality Generation.** For generating high-quality samples, we condition the model on maximum reward values across all N dimensions: $\hat{s}_{\max} = [B - 1, B - 1, \ldots, B - 1]$. This instructs the model to generate samples that maximize all reward objectives simultaneously.

**Multi-Reward Classifier-Free Guidance.** We extend classifier-free guidance to the multi-reward setting by leveraging the reward conditioning mechanism. Following the Coherence-Aware CFG approach (Dufour et al., 2024), we compute guidance using the contrast between a positive direction and a negative direction in the reward space. We introduce a *positive* and a *negative* reward target, denoted $\hat{s}^+$ and $\hat{s}^-$, which can be chosen by the user for controllability. By default, we use $\hat{s}^+ = \hat{s}_{\max} = [B - 1, \ldots, B - 1]$ and $\hat{s}^- = \hat{s}_{\min} = [0, \ldots, 0]$ and $\omega$ is the guidance scale:

$$\hat{v}_\theta(x_t, c) = v_\theta(x_t, c, \hat{s}^+) + \omega \left( v_\theta(x_t, c, \hat{s}^+) - v_\theta(x_t, c, \hat{s}^-) \right) \tag{3}$$

**Theoretical Interpretation.** This guidance formulation can be interpreted as approximating the gradient of an implicit joint reward function. Specifically, the guidance direction $v_\theta(x_t, c, \hat{s}_{\max}) - v_\theta(x_t, c, \hat{s}_{\min})$ points toward regions of the latent space where all rewards are simultaneously high, effectively steering generation away from low-quality outputs and toward samples that satisfy multiple

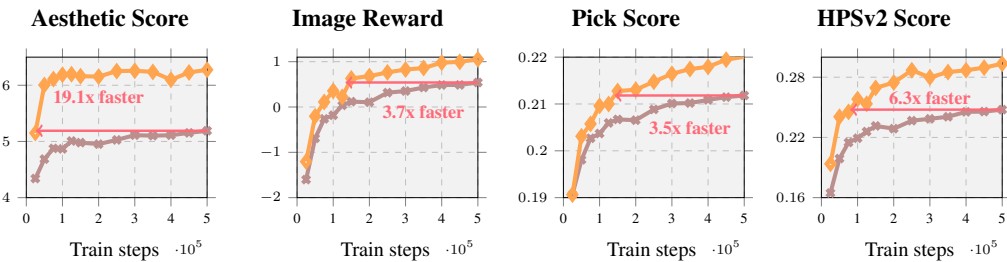

Figure 5: Training curves showing reward evolution during training. $\times$ Baseline, $\diamond$ MIRO.

quality criteria jointly. By amplifying this direction with the guidance scale $\omega$, we push the generated samples toward parts of the distribution characterized by superior aesthetic quality, text alignment, and other desired attributes. Similar to the weak guidance framework (Karras et al., 2024), where a bad version of the model is used to guide the good version, here the guidance is provided by the contrast between high-reward and low-reward conditioning.

**Flexible Reward Trade-offs.** A key advantage of MIRO is the ability to specify custom reward targets at inference time. Users can set $\hat{\mathbf{s}}_{\text{custom}} = [\hat{s}_1, \hat{s}_2, \ldots, \hat{s}_N]$ where each represents the desired level for reward $j$ for image $i$. This enables control over trade-offs between different quality or preference aspects.

## 2.4 ADVANTAGES OF MIRO OVER TRADITIONAL ALIGNMENT APPROACHES

MIRO offers several key advantages over traditional alignment approaches, stemming from its unified training paradigm and explicit reward conditioning mechanism.

**Training Efficiency.** By incorporating reward alignment directly into pretraining, MIRO eliminates the need for separate fine-tuning or reinforcement learning stages. MIRO converges to reward-aligned behavior without additional training phases achieving faster convergence than regular pretraining and higher quality samples. The single-stage training also reduces the complexity of the training pipeline and eliminates hyperparameter tuning for multiple stages.

**Full-Spectrum Data Utilization.** In contrast to post-hoc fine-tuning and RL pipelines that filter or concentrate training on a narrow slice of high-reward data, MIRO retains every sample and trains across the entire reward spectrum. Each example contributes signal together with its associated reward vector, so low-, medium-, and high-scoring regions are all modeled. This spectrum-wide supervision reduces collapse toward narrow high-reward modes, yields representations that generalize across quality levels, and produces a single model that can intentionally generate at any desired reward level at inference time.

**Reward Hacking Prevention.** Traditional single-objective optimization often leads to reward hacking, where models exploit specific reward metrics at the expense of overall quality (Luo et al., 2025). MIRO's multi-dimensional conditioning naturally prevents this by requiring the model to balance multiple objectives simultaneously. Users can detect and mitigate reward hacking by adjusting individual reward levels and observing the resulting trade-offs.

**Controllability and Interpretability.** The explicit reward conditioning provides interpretable control over generation quality. Users can understand and predict the effect of different reward settings, enabling more intuitive interaction with the model. This controllability extends beyond simple quality scaling to nuanced trade-offs between different aspects of visual quality.

## 3 EXPERIMENTS

### 3.1 REWARD-CONDITIONED PRETRAINING IMPROVES MODEL QUALITY

We demonstrate that pretraining with MIRO produces superior models compared to traditional approaches. We evaluate three training configurations: (1) a baseline model trained without reward conditioning, (2) single-reward models conditioned on individual rewards (similar to Coherence

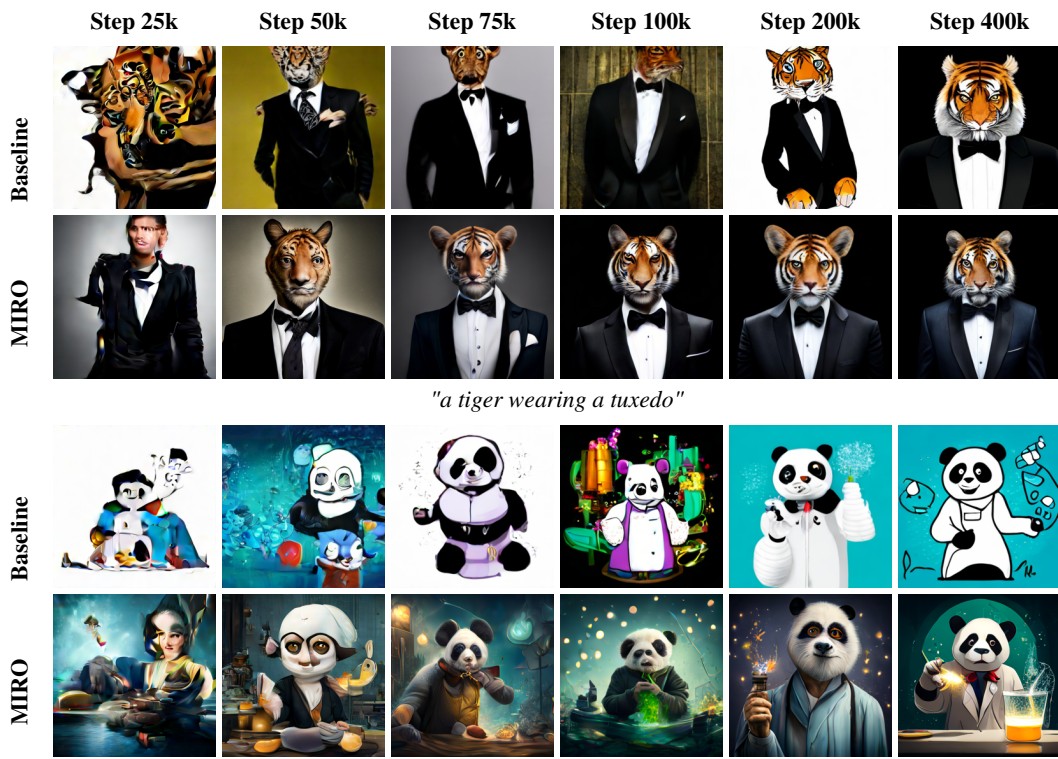

Figure 6: Training progression visualization showing generated images at different training steps for the same prompt. Top row shows baseline model outputs, bottom row shows MIRO model outputs.

Aware Diffusion Dufour et al. (2024) but using our reward suite instead of CLIP score), and (3) MIRO conditioned on all seven rewards simultaneously.

**MIRO outperforms single-reward approaches across all metrics.** Figure 4 presents results on the CC12M+LA6 dataset, evaluating models across AestheticScore, PickScore, ImageReward, HPSv2, and JINA CLIP score. We also include OpenAI CLIP score as an out-of-distribution evaluation metric not used during training. MIRO consistently outperforms all baselines across aesthetic and preference metrics, demonstrating the effectiveness of multi-reward conditioning.

**Multi-reward conditioning mitigates reward hacking.** Crucially, we observe that leveraging multiple rewards mitigates reward hacking compared to single-reward optimization. This is particularly evident with AestheticScore: while the single-reward model achieves high aesthetic scores, it severely degrades performance on other metrics. Models trained on ImageReward and HPSv2 show more balanced trade-offs but still underperform MIRO's comprehensive optimization.

**MIRO dramatically accelerates training convergence.** Figure 5 reveals substantial training efficiency gains from multi-reward conditioning. MIRO reaches the baseline model's final performance dramatically faster: 19× speedup for AestheticScore, 6.2× for HPSv2, 3.5× for PickScore, and 3.3× for ImageReward. This acceleration occurs because reward conditioning provides dense supervisory signals throughout training that guide the model toward high-quality generations, rather than requiring the model to discover these qualities through the diffusion objective alone.

**Qualitative results confirm accelerated high-quality generation.** Figure 6 provides qualitative evidence of MIRO's accelerated convergence. For the "tiger in a tuxedo" prompt, MIRO establishes proper compositional layout and generates a visually appealing tiger within 50k training steps—a level of quality that requires 200k steps for the baseline model to achieve. Similarly, for the "mad scientist panda" prompt, MIRO rapidly converges to aesthetically pleasing results while the baseline model fails to generate a recognizable panda until 400k steps. These qualitative improvements

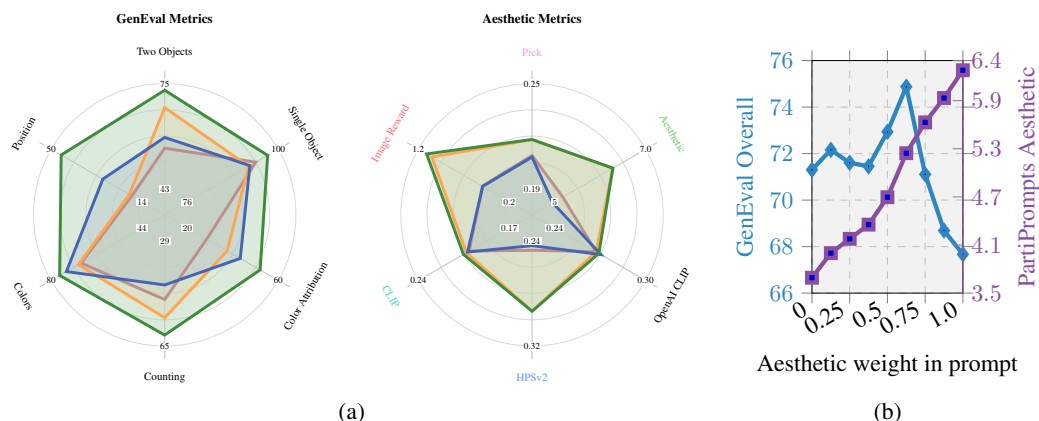

(a)  (b)

Figure 7: (a) MIRO vs baseline trained with real vs synthetic captions on GenEval and Aesthetic metrics. (b) Trading off GenEval and Aesthetic scores with Synth MIRO by adjusting the aesthetic weight in the prompt, i.e., varying the positive target $\hat{s}^+_{\text{aesthetic}}$ while keeping the other components of $\hat{s}^+$ equal to 1 and $\hat{s}^-$ fixed. Legend: **Baseline** (real), **MIRO** (real), **Synth Baseline** (50% real + 50% synth), **Synth MIRO** (50% real + 50% synth); lines in (b): GenEval Overall, Aesthetic.

complement our quantitative findings, demonstrating that MIRO's multi-reward conditioning enables both faster convergence and superior generation quality.

## 3.2 IMPROVING TEXT-IMAGE ALIGNMENT

Beyond optimizing for specific reward metrics, MIRO demonstrates significant improvements in text-image alignment, as measured by comprehensive evaluation benchmarks. Table 1 presents detailed results on GenEval, comparing MIRO against baseline models and single-reward approaches.

**MIRO enhances compositional understanding.** Our multi-reward approach substantially improves the model's ability to generate images that accurately reflect textual descriptions. MIRO achieves an overall GenEval score of 57, representing a 9.6% improvement over the baseline score of 52. This enhancement is particularly pronounced in challenging compositional reasoning tasks: Color Attribution improves from 29 to 38 (+31%), Two Objects from 55 to 68 (+24%), and Counting from 49 to 55 (+12%). These results demonstrate that MIRO's multi-reward conditioning enables better understanding of complex spatial relationships, object interactions, and numerical concepts.

**Single-reward models exhibit varying alignment capabilities.** Our analysis reveals that different reward models contribute differently to text-image alignment. Models optimized solely for aesthetic appeal (AestheticScore) achieve poor GenEval performance (33.0), suggesting that aesthetic optimization can come at the expense of semantic fidelity. In contrast, rewards more directly related to text-image correspondence—such as CLIP score, VQA score, and JINA CLIP score—achieve GenEval scores of 57, matching MIRO's performance. Notably, the SciScore model achieves the highest single-reward GenEval score of 58.0, though this comes with reduced aesthetic quality as shown in Figure 4.

**Multi-reward conditioning prevents overfitting.** The superior performance of MIRO compared to single-reward models highlights a key advantage of our approach: by optimizing across multiple complementary objectives simultaneously, MIRO avoids the overfitting that occurs when models focus exclusively on a single reward signal. This balanced optimization leads to models that excel across diverse evaluation criteria while maintaining strong performance on individual metrics.

## 3.3 MIRO AND SYNTHETIC CAPTIONS

Synthetic captioning has emerged as the go-to method for improving text-image alignment in generative models. This approach offers the advantage of retaining all training data without requiring filtering based on caption quality. While CAD Dufour et al. (2024) proposes a method to avoid

| Model | Params (B) | Inference TFLOPs | GenEval | | | | | | | PartiPrompts | | | |
|---|---|---|---|---|---|---|---|---|---|---|---|---|---|
| | | | Overall | Single Obj. | Two Obj. | Position | Counting | Colors | Color Attr. | Aesthetic | Image | HPSv2 | PickAScore |
| *SOTA Baselines* | | | | | | | | | | | | | |
| SD v1.5 | 0.9 | - | 43 | 97 | 38 | 4 | 35 | 76 | 6 | 5.68 | 0.24 | 0.25 | 0.213 |
| SD v2.1 | 0.9 | - | 50 | 98 | 51 | 7 | 44 | 85 | 17 | 5.81 | 0.38 | 0.26 | 0.215 |
| PixArt-$\alpha$ | 0.6 | - | 48 | 98 | 50 | 8 | 44 | 80 | 7 | 6.47 | 0.97 | 0.29 | 0.226 |
| PixArt-$\Sigma$ | 0.6 | - | 52 | 98 | 59 | 10 | 50 | 80 | 15 | 6.44 | 1.02 | 0.29 | 0.225 |
| CAD | 0.35 | 20.8 | 50 | 95 | 56 | 11 | 40 | 76 | 22 | 5.56 | 0.69 | 0.26 | 0.214 |
| Sana-0.6B | 0.6 | - | 64 | 99 | 71 | 16 | 63 | 91 | 42 | 6.31 | 1.23 | 0.30 | 0.228 |
| Sana-1.6B | 1.6 | - | 66 | 99 | 79 | 18 | 63 | 88 | 47 | 6.36 | 1.23 | 0.30 | 0.228 |
| SDXL | 2.6 | - | 55 | 98 | 74 | 15 | 39 | 85 | 23 | 5.94 | 0.46 | 0.25 | 0.220 |
| SD3-medium | 2.0 | - | 62 | 98 | 74 | 34 | 67 | 67 | 36 | 6.18 | 1.15 | 0.30 | 0.225 |
| FLUX-dev | 12.0 | 1540 | 67 | 99 | 81 | 20 | 79 | 74 | 47 | 6.56 | 1.19 | 0.30 | 0.229 |
| *CAD-like Models (our models)* | | | | | | | | | | | | | |
| Image Reward | 0.36 | 4.16 | 57 | 97 | 59 | 21 | 56 | 76 | 33 | 5.31 | 1.04 | 0.27 | 0.214 |
| HPSv2 | 0.36 | 4.16 | 56 | 95 | 63 | 15 | 52 | 78 | 31 | 5.47 | 0.90 | 0.29 | 0.215 |
| Aesthetic | 0.36 | 4.16 | 33 | 74 | 37 | 6 | 24 | 42 | 15 | 6.65 | 0.00 | 0.26 | 0.209 |
| SciScore | 0.36 | 4.16 | 58 | 94 | 62 | 24 | 61 | 72 | 35 | 4.62 | 0.56 | 0.24 | 0.209 |
| CLIP | 0.36 | 4.16 | 57 | 97 | 63 | 24 | 57 | 70 | 32 | 5.04 | 0.73 | 0.25 | 0.214 |
| VQA | 0.36 | 4.16 | 57 | 97 | 58 | 20 | 57 | 76 | 37 | 4.88 | 0.64 | 0.25 | 0.212 |
| Pick | 0.36 | 4.16 | 57 | 93 | 62 | 17 | 58 | 75 | 34 | 5.16 | 0.76 | 0.26 | 0.216 |
| *Real Caption Models (our models)* | | | | | | | | | | | | | |
| Baseline | 0.36 | 4.16 | 52 | 94 | 55 | 18 | 49 | 68 | 29 | 5.18 | 0.52 | 0.25 | 0.212 |
| MIRO | 0.36 | 4.16 | 57 | 92 | 68 | 19 | 55 | 69 | 38 | 6.28 | 1.06 | 0.29 | 0.220 |
| *Synthetic Caption Models (50% Real + 50% Synth) (our models)* | | | | | | | | | | | | | |
| Baseline | 0.36 | 4.16 | 57 | 93 | 59 | 30 | 44 | 74 | 43 | 4.96 | 0.52 | 0.24 | 0.211 |
| MIRO | 0.36 | 4.16 | 68 | 97 | 73 | 46 | 61 | 77 | 52 | 6.28 | 1.11 | 0.29 | 0.220 |
| MIRO† | 0.36 | 4.16 | 75 | 98 | 79 | 58 | 71 | 85 | 58 | 5.24 | 1.18 | 0.29 | 0.220 |
| *Inference Scaled + Synthetic Caption Models (MIRO + 128 samples inference scaled) (our models)* | | | | | | | | | | | | | |
| Aesthetic Scaled MIRO | 0.36 | 532 | 63 | 97 | 68 | 40 | 57 | 75 | 45 | 6.81 | 1.04 | 0.29 | 0.219 |
| Image Reward Scaled MIRO | 0.36 | 532 | 75 | 98 | 84 | 52 | 69 | 82 | 65 | 6.28 | 1.61 | 0.30 | 0.223 |
| HPSv2 Scaled MIRO | 0.36 | 532 | 74 | 98 | 83 | 47 | 74 | 80 | 65 | 6.28 | 1.35 | 0.32 | 0.225 |
| PickAScore Scaled MIRO | 0.36 | 532 | 74 | 98 | 83 | 44 | 76 | 81 | 59 | 6.27 | 1.32 | 0.31 | 0.229 |

Table 1: GenEval and PartiPrompts Results Comparison Across All Models. Unless noted, inference uses the positive/negative targets $\hat{s}^+ = [1, 1, \ldots, 1]$ and $\hat{s}^- = [0, 0, \ldots, 0]$. † denotes a custom positive target with all rewards set to 1 except the aesthetic reward set to 0.625 (i.e., $\hat{s}^+_{\text{aesthetic}} = 0.625$), with $\hat{s}^-$ fixed.

filtering, it does not demonstrate results on synthetic captions. We evaluate MIRO using a mixture of 50% synthetic and 50% real captions (captioning details provided in the Supplementary Material).

**Technical implementation.** Applying MIRO to synthetic captions presents a challenge: some reward models cannot process captions longer than 77 tokens, while our synthetic captions are extensive (approximately 200 tokens). To address this limitation, we generate both long synthetic captions for training and shorter versions for reward model evaluation.

**MIRO outperforms synthetic captioning alone.** Our results demonstrate that MIRO without synthetic captions achieves comparable GenEval performance to baseline models trained with synthetic captions. More importantly, Figure 7a shows that MIRO without synthetic captions significantly outperforms the synthetic caption baseline across rewards metrics. This finding suggests that MIRO provides a more effective approach to improving text-image alignment than synthetic captioning alone, while being computationally more efficient. Indeed, reward model scoring requires substantially less compute than recaptioning with large vision-language models.

**MIRO unlocks synthetic caption potential.** Combining MIRO with synthetic captions yields the strongest overall performance as shown in Table 1. While maintaining equivalent aesthetic quality to MIRO without synthetic captions, this combined approach achieves a remarkable GenEval score of 68, substantially improving over the synthetic caption baseline of 57 (+19%). The improvements are consistent across all compositional reasoning metrics: Position increases from 30 to 46 (+53%), Color Attribution from 43 to 52 (+21%), Single Object from 93 to 97 (+4%), Two Objects from 58 to 73 (+26%), and Counting from 44 to 61 (+39%). These comprehensive gains across all compositional aspects demonstrate that MIRO effectively benefits massively from synthetic captions for text-image alignment, achieving superior compositional understanding while preserving aesthetic quality.

## 3.4 Synergizing with Test-Time Scaling

Test-time scaling has emerged as a popular method to improve reward performance by generating multiple samples and selecting the best one Ma et al. (2025). We demonstrate that MIRO achieves superior sample efficiency compared to baseline models when combined with test-time scaling.

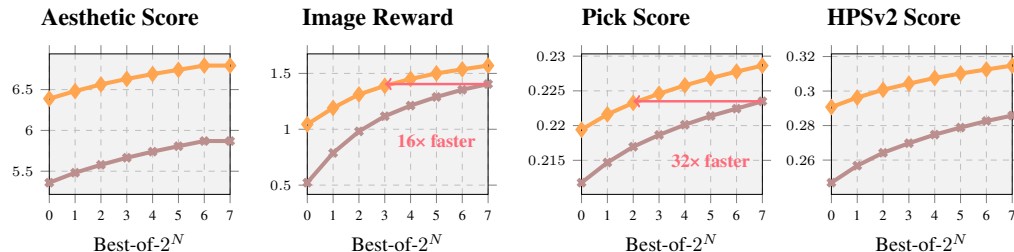

Figure 8: Test-time scaling showing performance vs. Best-of-$2^N$ sampling. $\times$ Baseline, $\diamond$ MIRO.

**Experimental setup.** We evaluate both baseline and MIRO models using the Random Search protocol from Ma et al. (2025). Figure 8 presents performance across varying sample counts (1 to 128 samples, displayed on a log-2 scale). For each evaluation, we generate N samples and select the highest-scoring sample according to the respective reward model.

**MIRO demonstrates superior sample efficiency.** Our results reveal that MIRO consistently outperforms the baseline across all reward metrics, often by substantial margins. Most remarkably, for Aesthetic Score and HPSv2 metrics, MIRO achieves with a single sample what the baseline cannot reach even with 128 samples. This dramatic efficiency gain highlights MIRO's ability to generate high-quality samples without requiring extensive test-time computation.

**Quantifying inference-time efficiency improvements.** The efficiency gains are particularly striking for specific metrics: For ImageReward, MIRO with 8 samples matches the performance of the baseline with 128 samples, representing a 16× efficiency improvement. For PickScore, MIRO achieves equivalent performance with only 4 samples compared to the baseline's 128 samples, demonstrating a remarkable 32× efficiency gain. These results establish MIRO as not only a superior training approach but also a more efficient inference-time method.

### 3.5 COMPARISON TO STATE-OF-THE-ART MODELS

In Figure 1, we evaluate MIRO against state-of-the-art text-to-image models on GenEval, demonstrating superior performance while maintaining significantly lower computational costs.

**GenEval results demonstrate exceptional training efficiency.** MIRO achieves a GenEval score of 68, outperforming FLUX-dev (12B parameters) which scores 67, while requiring dramatically less computation: 4.16 TFLOPs vs 1540 TFLOPs for FLUX-dev, representing a remarkable 370× efficiency improvement. This demonstrates that MIRO's multi-reward conditioning enables compact models to surpass much larger architectures.

**MIRO sets new benchmarks for compositional reasoning.** Beyond overall performance, MIRO excels on challenging compositional metrics that have historically been difficult for text-to-image models. On the Position metric, MIRO achieves a score of 46, improving upon the previous state-of-the-art of 34 (SD3-Medium) by 31%. For Color Attribution, MIRO advances from FLUX-dev's previous best of 47 to 52 (+11%).

**User preference evaluation confirms scalable efficiency.** On PartiPrompts, MIRO consistently outperforms larger models across multiple reward metrics, leveraging inference time scaling. When optimizing for Aesthetic Score with 128-sample inference scaling, MIRO achieves a state-of-the-art score of 6.81 compared to FLUX-dev's 6.56. For ImageReward optimization, MIRO scores 1.61 versus Sana-1.6B's 1.23. Remarkably, even with this 128-sample inference scaling strategy, MIRO maintains a 3× efficiency advantage over FLUX-dev (532 TFLOPs vs 1540 TFLOPs) while achieving superior performance across all metrics.

**Multi-reward conditioning enables cross-metric generalization.** Notably, MIRO demonstrates strong performance even when not explicitly optimized for specific metrics. For instance, when optimizing for HPSv2, MIRO achieves an ImageReward score of 1.35, outperforming models specifically trained for that metric. This cross-metric generalization highlights the robustness of MIRO's multi-reward approach and its ability to achieve state-of-the-art results with substantially reduced computational requirements.

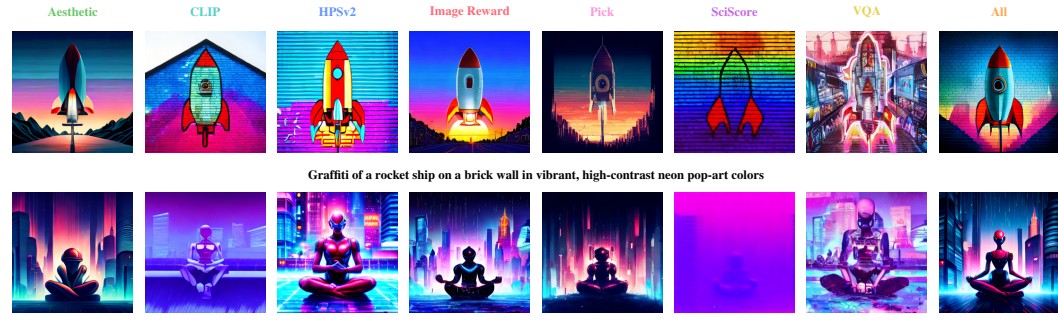

Figure 9: **Generations from the Synth MIRO model using multi-reward classifier-free guidance** (see notation in Section 2.3). For each column $j$, we sample with a positive target $\hat{\mathbf{s}}^+ = [1, \ldots, 1]$ and a negative target $\hat{\mathbf{s}}^- = [1, \ldots, 1]$ except $\hat{s}_j^- = 0$. The guidance vector points purely toward reward $j$. The "All" column uses $\hat{\mathbf{s}}^- = \mathbf{0}$, guiding toward simultaneously high values for all rewards.

### 3.6 FLEXIBLE REWARD TRADE-OFFS AT INFERENCE

**Reward weighting exposes controllable trade-offs between aesthetics and alignment.** Our test-time scaling results (Figure 8) show that selecting samples by Aesthetic Score can reduce GenEval performance, indicating a trade-off between aesthetic quality and semantic alignment.

**Sweeping the aesthetic weight identifies an optimal balance.** We vary the aesthetic reward weight at inference and observe the highest GenEval score at a weight of 0.625 (Figure 12).

**Optimized weighting rivals heavy test-time scaling.** Using this inference strategy, MIRO† match the GenEval performance of ImageReward-based selection with 128-sample test-time scaling, while using a single weighted selection. Other metrics also improve; for example, ImageReward reaches 1.18, matching FLUX-dev without test-time scaling.

**Visualizing per-reward controllability.** In Figure 9, we visualize this controllability with Synth MIRO using multi-reward classifier-free guidance (Section 2.3). For column $j$, we set $\hat{\mathbf{s}}^+ = [1, \ldots, 1]$ and $\hat{\mathbf{s}}^- = [1, \ldots, 1]$ with $\hat{s}_j^- = 0$, which cancels the shared direction and isolates reward $j$ while keeping the other rewards anchored to $\hat{\mathbf{s}}^+$.

**Pairwise reward exploration.** To explore the trade-offs between two specific rewards, we perform pairwise interpolation while keeping all other rewards fixed. For rewards $A$ and $B$, we set $\hat{\mathbf{s}}^+ = [1, \ldots, 1]$ and $\hat{\mathbf{s}}^- = [1, \ldots, 1]$, except for the two rewards of interest: $\hat{s}_A^- = t$ and $\hat{s}_B^- = 1 - t$, where $t \in [0, 1]$ controls the interpolation. This configuration enables smooth exploration of the reward space between two objectives while maintaining high values for all other rewards, revealing the model's ability to navigate trade-offs between specific quality dimensions.

**User-controlled rewards at inference.** MIRO allows choosing reward weights at test time, enabling principled trade-offs across capabilities, giving users control and reducing reward hacking.

## 4 CONCLUSION

We presented Multi-Reward cOnditioning (**MIRO**), a simple pretraining framework that conditions on a vector of reward scores to integrate alignment into training rather than as a post-hoc stage. By learning $p(x \mid c, \mathbf{s})$ and exposing reward targets as controllable inputs, **MIRO** disentangles content from quality, offering precise and interpretable control at inference time. Empirically, on a 16M-image setup, **MIRO** outperforms no-conditioning and single-reward baselines, converges substantially faster, mitigates reward hacking, strengthens compositional alignment, and achieves state-of-the-art results on PartiPrompts with inference-time scaling, while being markedly more compute-efficient. Notably, despite being much smaller, our **MIRO** model surpasses FLUX-dev on GenEval and PartiPrompts at a fraction of the compute. We hope that this work will pave the way for this alternative line of research on how to exploit rewards at pre-training.

## REPRODUCIBILITY STATEMENT

To ensure the reproducibility of our results, we have provided a comprehensive description of the MIRO framework and its implementation. The complete source code for dataset augmentation, pretraining, and the multi-reward inference pipeline is available here <ULR>. Detailed specifications of the model architecture and the flow matching objective are outlined in Section 2 and Appendix B. We utilized publicly available datasets (CC12M and LAION Aesthetics 6+), and the specific synthetic captioning pipeline is detailed in Section 3.3 and Appendix E. The exact configurations for the seven reward models, along with the score normalization and binning strategies, are documented in Section 2.1 and Appendix B, It allows the exact replication of our training environment and experimental baselines.

## ETHIC STATEMENT

**Data and Privacy** We utilized publicly available datasets: CC12M and LAION Aesthetics 6+. While these datasets are standard in the field, we acknowledge the ongoing community discussions regarding the presence of copyrighted material and private individuals in web-crawled data.

**Bias and Fairness** MIRO conditions generation on reward models such as HPSv2, PickScore, and Aesthetic Score. We caution that these reward models reflect the preferences and biases of the specific user groups that annotated their training data. Optimizing for "high reward" may inadvertently amplify societal biases regarding beauty standards, race, or gender roles. Users should be aware that the "quality" defined by these rewards is subjective and not culturally universal.

**Environmental Impact** A core contribution of this work is efficiency. MIRO converges up to $19\times$ faster than baseline pretraining and requires orders of magnitude less inference compute (e.g., $370\times$ less than FLUX-dev) to achieve comparable quality. We believe this direction significantly contributes to reducing the carbon footprint associated with training and deploying high-quality generative models.

**Misuse** As with all open-domain text-to-image models, there is a risk of generating harmful, offensive, or misleading content. While multi-reward conditioning improves alignment with safe prompts, it does not inherently prevent the generation of malicious content if explicitly prompted.

## THE USE OF LLMs

We acknowledge the use of Large Language Models to assist in the preparation of this manuscript. The usage was limited to polishing the writing, refining sentence structure, and correcting grammatical errors. The research conceptualization and experimental design were performed without LLM assistance.

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

## A   RELATED WORK

### A.1   DIFFUSION, FLOW MATCHING, AND CONDITIONAL GENERATION

Modern T2I builds on diffusion/score models (Sohl-Dickstein et al., 2015; Ho et al., 2020; Song et al., 2021; Song & Ermon, 2019; 2020; Dhariwal & Nichol, 2021) and their latent and text-conditional variants (Rombach et al., 2022; Saharia et al., 2022; Ramesh et al., 2022). Transformer-based diffusion backbones (Peebles & Xie, 2023; Ma et al., 2024) and flow matching (Lipman et al., 2023) further improve scalability and training dynamics. Alternative backbones and simplifications broaden design space (Peebles & Xie, 2023; Hoogeboom et al., 2023; Jabri et al., 2023), while control modules and training/sampling variants provide additional levers (Zhang et al., 2023; Song et al., 2023; Luo et al., 2023). Practical recipes add stable normalization/activation and attention scaling (Zhang & Sennrich, 2019; Shazeer, 2020; Henry et al., 2020). Coherence-aware conditioning improves conditional generation without discarding data (Dufour et al., 2024) and Ambient Diffusion Omni (Daras et al., 2025) improves training with bad data.

### A.2   EFFICIENT TEXT-TO-IMAGE GENERATION

Efficiency arises from data, objectives, and architectures. Compact, public datasets enable reproducible training with lower compute (Changpinyo et al., 2021; Schuhmann et al., 2022; Thomee et al., 2016; Gokaslan et al., 2024; Deng et al., 2009; Degeorge et al., 2025). Representation-focused objectives and recipes accelerate convergence (Wei et al., 2023; Yu et al., 2024). Transformer-based diffusion/flow models (Peebles & Xie, 2023; Ma et al., 2024) and latent training (Rombach et al., 2022; Chen et al., 2024b; Gu et al., 2023) reduce cost while preserving quality; large-scale systems highlight the upper bound in capability and compute (Betker et al., 2023; Esser et al., 2024). Coherence-aware training further improves sample efficiency without filtering (Dufour et al., 2024).

### A.3   ALIGNING T2I MODELS WITH REWARD SIGNALS AND TEST-TIME SCALING

Reward models span complementary axes for alignment and evaluation (AestheticScore, HPSv2, ImageReward, PickScore, VQAScore, JINA CLIP, SciScore) (Schuhmann et al., 2022; Wu et al., 2023; Xu et al., 2023; Kirstain et al., 2023; Lin et al., 2024; Koukounas et al., 2024; Li et al., 2025). Training-time alignment either fine-tunes diffusion models with reward feedback via RL (Black et al., 2024; Fan et al., 2023; Deng et al., 2025)—effective but compute-heavy and sometimes unstable—or learns from preferences using pairwise objectives such as DPO (Rafailov et al., 2023; Wallace et al., 2024; Li et al., 2024). Control-theoretic formulations optimize continuous-time dynamics with reward guidance (Uehara et al., 2024; Tang & Zhou, 2025; Domingo-Enrich et al., 2025) but are costly; lighter approaches avoid full trajectory gradients (Oertell et al., 2024; Miao et al., 2024; Jia et al., 2024). At inference, test-time scaling boosts rewards via sample-and-select (Ma et al., 2025; Uehara et al., 2025) or reward-guided refinement (Ben-Hamu et al., 2024; Tang et al., 2024), at higher runtime cost. Complementary gradient-based alignment optimizes the initial noise using reward gradients (ReNO) and amortizes such test-time compute via Noise Hypernetworks (Eyring et al., 2024; 2025). MIRO conditions on multiple rewards during pretraining, enabling controllable trade-offs and strong single-sample quality, and complements test-time scaling by achieving higher scores with fewer samples and better GenEval alignment (Ghosh et al., 2024). Trading-off multiple rewards during inference has been explored by weight averaging methods like Rewarded Soups (Rame et al., 2023) but this approaches requires 1 model per reward, making it impractical for large number of rewards. Furthermore, changing the mix of rewards at inference time requires changing the model averaging parameters, which requires having all the models in memory.

## B   IMPLEMENTATION DETAILS

**Architecture Modifications.** Our flow matching network $v_\theta$ takes as input the noisy sample $x_t$, text condition $c$, and the binned reward vector $\hat{\mathbf{s}} = [\hat{s}_1, \hat{s}_2, \ldots, \hat{s}_N]$. The reward conditioning is implemented through:

- **Sinusoidal embeddings**: Each reward bin index $\hat{s}_i$ is encoded using sinusoidal position embeddings, similar to those used in transformer architectures

- **Token space mapping**: The sinusoidal reward embeddings are projected to the same dimensional space as text tokens
- **Token concatenation**: The projected reward embeddings are concatenated to the text token sequence, allowing the model to process rewards and text through the same attention mechanism

**Rewards preprocessing** For each reward model $r_j$, we:

1. Compute scores on a representative subset $\mathcal{D}_{\text{cal}} \subset \mathcal{D}$ of the training data
2. Sort the scores and divide them into $B$ bins with equal population
3. Map each raw score $s_j^{(i)}$ to its corresponding bin index $\hat{s}_j^{(i)} \in \{0, 1, \ldots, B-1\}$

This binning approach provides several advantages: (1) it normalizes different reward scales into a common discrete space, (2) ensures balanced training across all quality levels, and (3) provides interpretable conditioning signals where higher bin indices correspond to better quality.

**Experimental Setup** We used the TextRIN architecture Dufour et al. (2024) with several modifications: FFN layers replaced with SwiGLU Shazeer (2020), LayerNorm replaced with RM-SNorm Zhang & Sennrich (2019), and QK-Norm Henry et al. (2020) in attention mechanisms. We employed flow matching instead of diffusion for generation. Models were trained for 500k steps with batch size 1,024 and learning rate 1e-3. We train our model in 256px resolution. We combined CC12M Changpinyo et al. (2021) and LAION Aesthetics 6+ Schuhmann et al. (2022) for 16M total image-text pairs, following Dufour et al. (2024). We used seven reward models for MIRO: **Aesthetic Score** Schuhmann et al. (2022) for visual appeal, **HPSv2** Wu et al. (2023) for human preference alignment, **ImageReward** Xu et al. (2023) for text-image correspondence and user preference, **PickScore** Kirstain et al. (2023) for user preference, **VQAScore** Lin et al. (2024) for visual comprehension, **JINA CLIP Score** Koukounas et al. (2024) for long captions CLIP score, and **SciScore** Li et al. (2025) for scientific accuracy.

**SOTA Baselines** We compare MIRO against the following baselines:

- **SD v1.5**: Rombach et al. (2022)
- **SD v2.1**: Rombach et al. (2022)
- **PixArt-$\alpha$**: Chen et al. (2024b)
- **PixArt-$\Sigma$**: Chen et al. (2024a)
- **CAD**: Dufour et al. (2024)
- **Sana-0.6B**: Xie et al. (2024)
- **Sana-1.6B**: Xie et al. (2024)
- **SDXL**: Podell et al. (2024)
- **FLUX-dev**: Labs (2024)
- **SD3-Medium**: Esser et al. (2024)

# C ADDITIONAL RESULTS

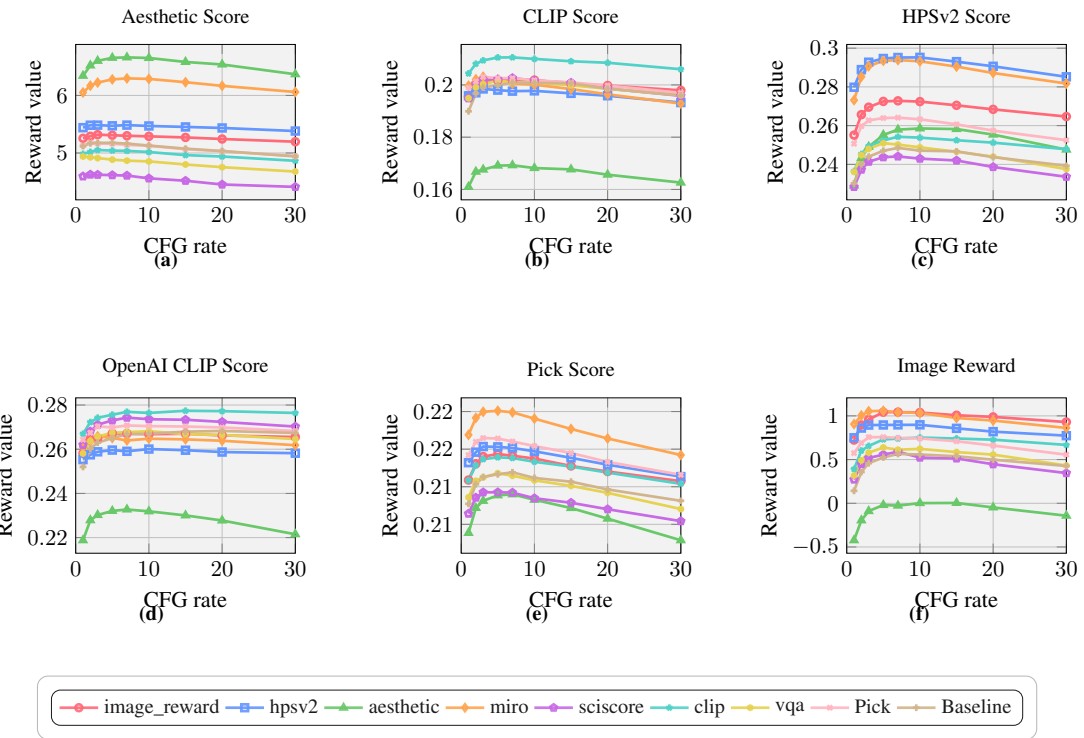

Figure 10: Score plots for different reward functions: (a) Aesthetic Score, (b) CLIP Score, (c) HPSv2 Score, (d) OpenAI CLIP Score, (e) Pick Score, and (f) Image Reward. Each plot shows all models color-coded according to the legend.

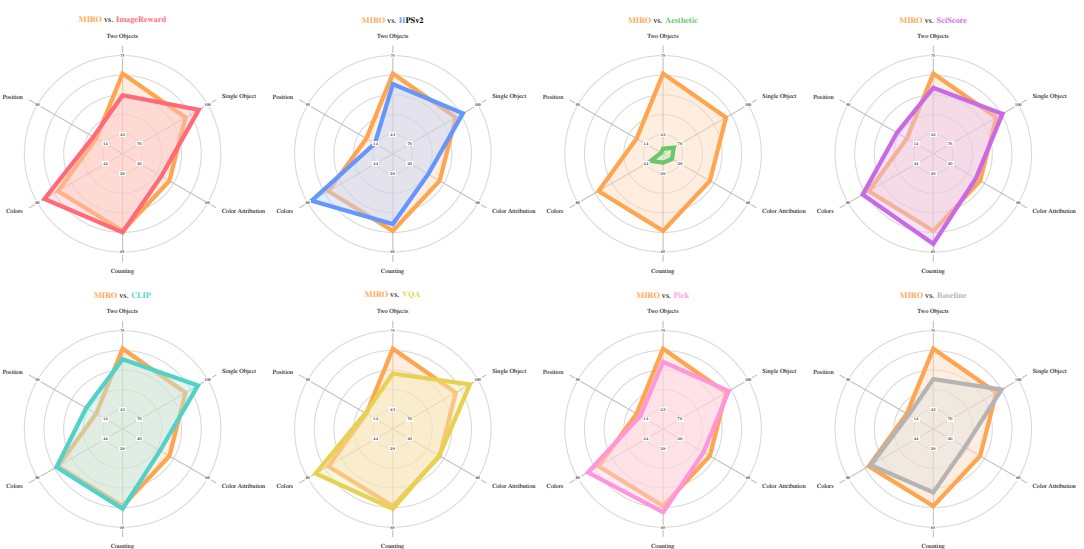

Figure 11: Comparison of the MIRO model against eight other specialist/baseline models on GenEval metrics. Each radar plot shows the MIRO model (orange) versus a comparison model across six GenEval categories: Single Object, Two Objects, Position, Counting, Colors, and Color Attribution. Scores range from 0 to 100 for all categories. Min and max values on each axis show the range of actual metric scores and are consistent across all plots.

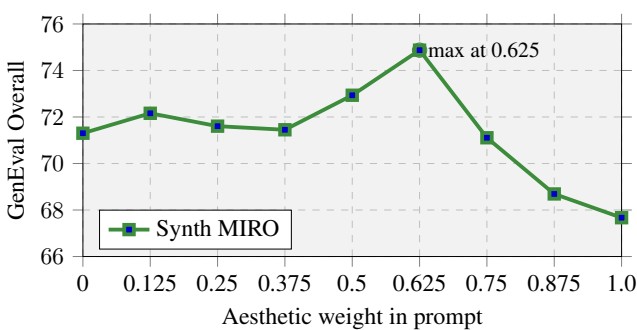

Figure 12: GenEval Overall vs aesthetic prompt weight for 'Synth MIRO'. We vary the positive target $\hat{s}^+_{\text{aesthetic}}$ while keeping the other components of $\hat{s}^+$ equal to 1 and $\hat{s}^-$ fixed. Higher is better.

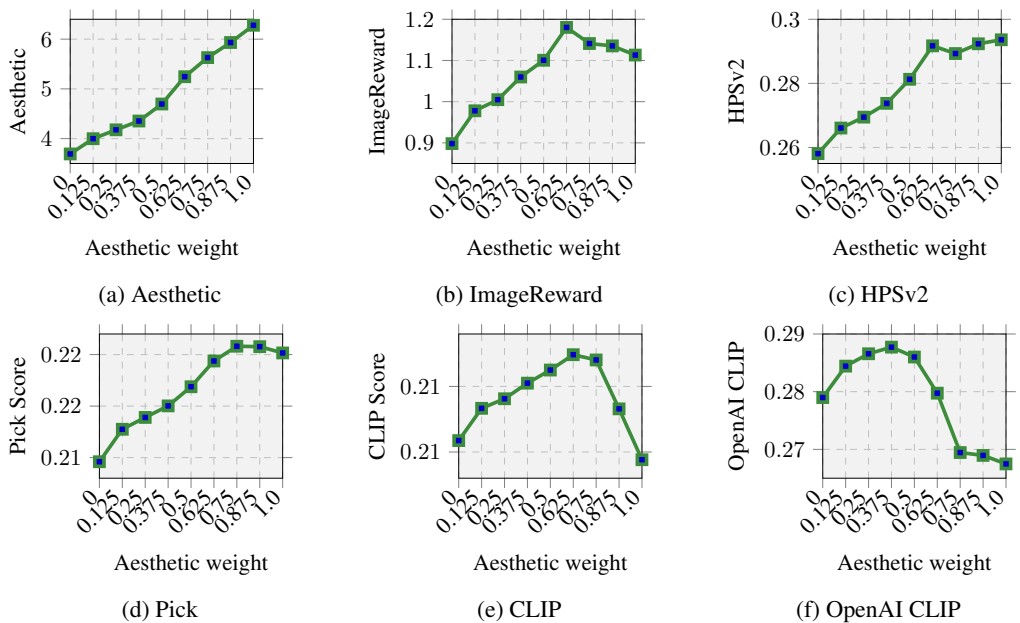

(a) Aesthetic        (b) ImageReward        (c) HPSv2

(d) Pick        (e) CLIP        (f) OpenAI CLIP

Figure 13: Six metrics vs aesthetic prompt weight for 'Synth MIRO'. Each subplot shows the metric value over aesthetic weight in the prompt.

## C.1 LEAVE-ONE-OUT EXPERIMENTS

We analyze the contribution of each reward model by training MIRO with one reward removed at a time. Figure 15 shows the impact on reward metrics, while Figure 16 shows the impact on Geneval metrics.

**Analysis:** As expected, removing a specific reward generally leads to a decrease in that specific metric. For instance, removing the **Aesthetic Score** leads to a significant drop in the aesthetic metric (from 6.23 to 5.05). However, interestingly, this removal leads to substantial improvements in Geneval metrics, particularly for **Position** (+11.8 points) and **Two Objects** (+11.4 points). This suggests a potential trade-off between optimizing for pure visual aesthetics and maintaining strict semantic fidelity or spatial composition. Conversely, removing **ImageReward** or **CLIP** tends to hurt semantic metrics like Two Object detection and Color accuracy, highlighting their importance for text-image alignment.

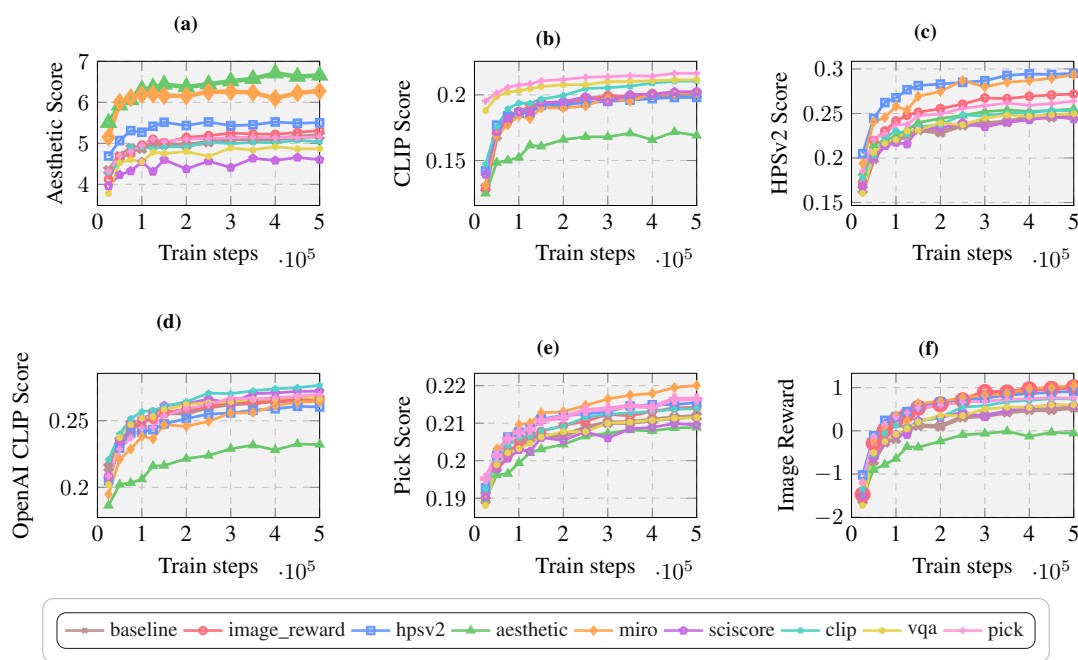

Figure 14: Training curves for different reward functions: (a) Aesthetic Score, (b) CLIP Score, (c) HPSv2 Score, (d) OpenAI CLIP Score, (e) Pick Score, and (f) Image Reward. Each plot shows the reward value progression across Train steps for different models including image_reward, hpsv2, aesthetic, miro, sciscore, clip, vqa, and pick.

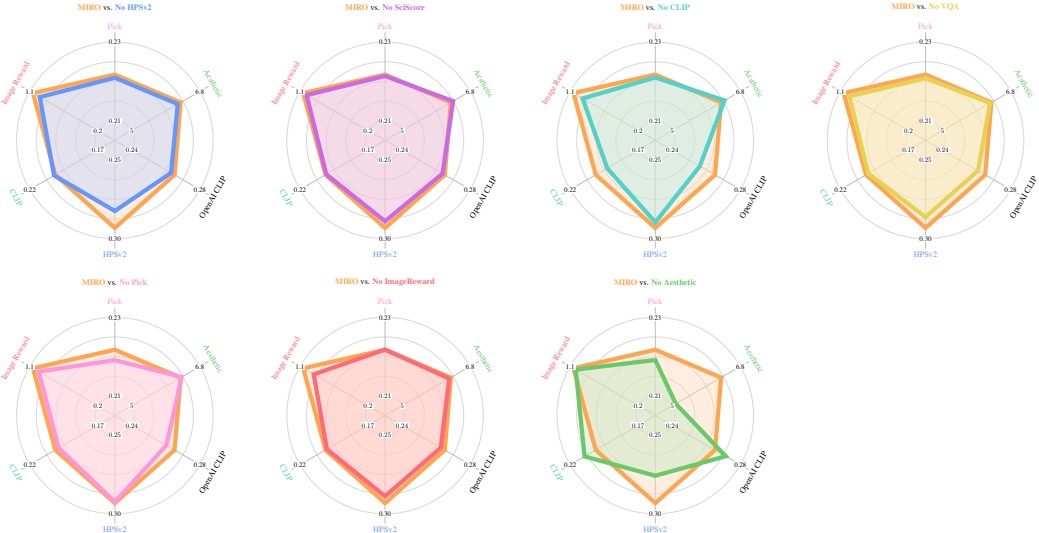

Figure 15: Ablation study: MIRO (Quantile Bins) vs MIRO leaving one reward out.

## C.2 BINNING ABLATION

We compare three binning strategies for the reward conditioning: **Quantile Bins** (our default MIRO), **Refined Quantile Bins** (64 quantiles with the last 8 re-binned), and **Uniform Bins**.

**Analysis:** The **Refined Quantile** strategy generally achieves the highest reward scores (e.g., Aesthetic 6.40 vs 6.23 for Quantile), indicating that finer granularity at the top end of the reward distribution helps the model target high-quality outputs. On Geneval, the results are mixed: Refined Quantile improves **Counting** and **Two Object** detection compared to standard Quantile bins. However,

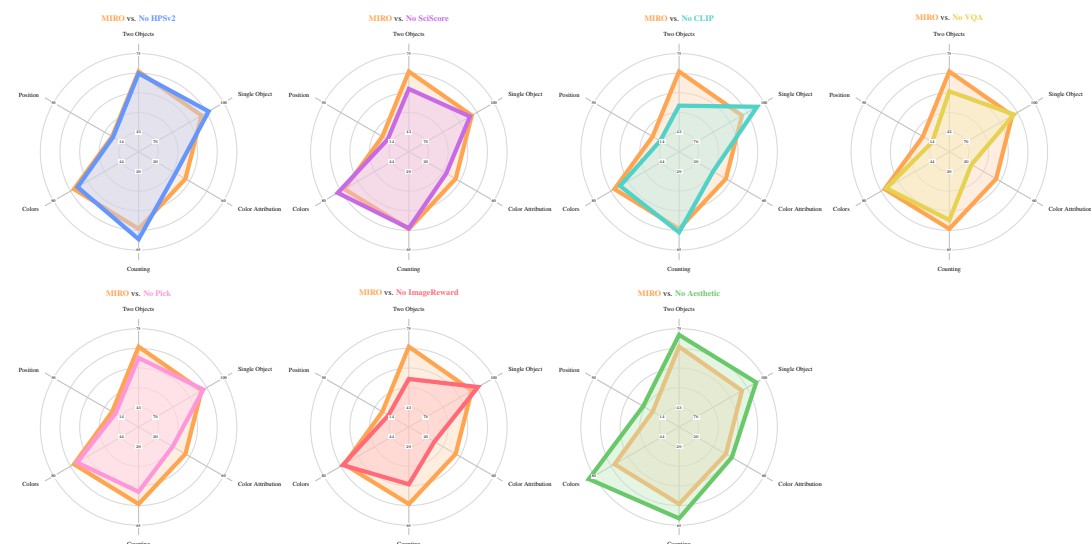

Figure 16: Ablation study on GenEval: MIRO vs MIRO leaving one reward out.

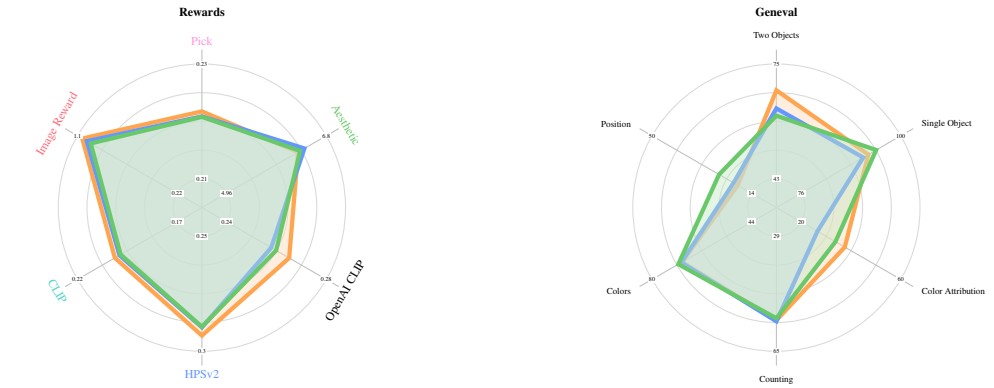

Figure 17: Comparison of **MIRO**, **Refined**, and **Uniform** bins on Rewards (Left) and Geneval (Right).

**Uniform Bins** perform surprisingly well on the **Position** metric (25.8 vs 12.2 for Quantile), suggesting that a uniform spread of conditioning signals might be beneficial for learning spatial relationships, possibly by providing more distinct signals across the entire quality spectrum.

## C.3 POST-TRAINING ABLATION

We investigate the efficiency of MIRO by comparing a model trained from scratch with MIRO (500k steps) against a **Post-training** approach, where a baseline model trained for 450k steps is fine-tuned with MIRO for only 50k steps.

**Analysis:** The Post-training approach demonstrates remarkable efficiency. Despite being trained with MIRO for only 10% of the total steps, it matches the full MIRO model on key reward metrics like **Aesthetic Score** (6.22 vs 6.23) and **HPSv2**. Even more notably, the Post-training model outperforms the full MIRO model on several Geneval metrics, particularly **Position** (23.2 vs 12.2) and **Single Object** detection. This suggests that initializing with a strong baseline and then fine-tuning with MIRO is not only more efficient but may also preserve some of the semantic knowledge (like spatial understanding) that can be destabilized during long multi-objective training from scratch.

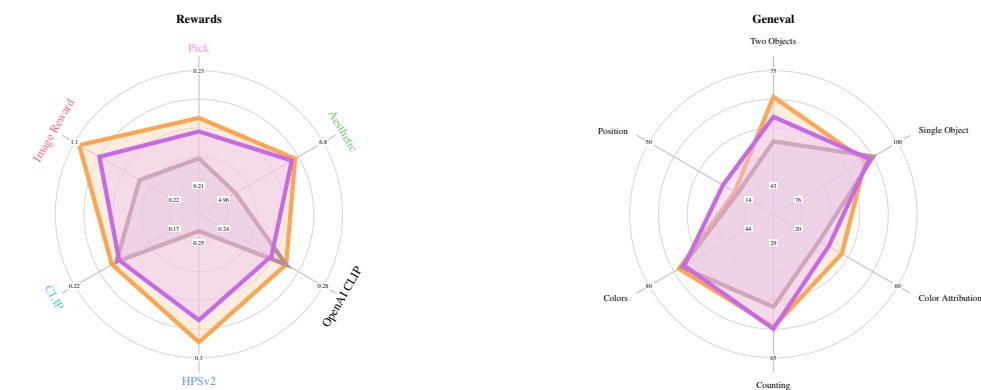

Figure 18: Comparison of **MIRO**, **Baseline**, and **Post-Train** on Rewards (Left) and Geneval (Right).

| Model | GenEval | | | | | | | PartiPrompts | | | |
|---|---|---|---|---|---|---|---|---|---|---|---|
| | Overall | Single Obj. | Two Obj. | Position | Counting | Colors | Color Attr. | Aesthetic | Image | HPSv2 | PickAScore |
| *Reference* | | | | | | | | | | | |
| Baseline | 52 | 94 | 55 | 18 | 49 | 68 | 29 | 5.18 | 0.52 | 0.25 | 0.212 |
| MIRO | 57 | 92 | 68 | 19 | 55 | 69 | 38 | 6.28 | **1.06** | **0.29** | **0.220** |
| *Leave-one-out Ablations* | | | | | | | | | | | |
| No Aesthetic | **63** | **97** | **72** | 24 | **62** | **83** | **41** | 5.05 | 1.03 | 0.28 | 0.217 |
| No Pick | 53 | 93 | 63 | 17 | 50 | 68 | 30 | 6.31 | 0.98 | **0.29** | 0.217 |
| No ImageReward | 51 | 94 | 55 | 16 | 46 | 70 | 25 | 6.23 | 0.92 | **0.29** | **0.220** |
| No HPSv2 | 57 | 95 | 67 | 18 | 60 | 67 | 32 | 6.20 | 0.97 | 0.28 | 0.219 |
| No CLIP | 53 | **98** | 54 | 15 | 57 | 66 | 30 | 6.37 | 0.94 | **0.29** | 0.219 |
| No SciScore | 55 | 92 | 61 | 16 | 55 | 73 | 32 | 6.34 | 1.01 | **0.29** | **0.220** |
| No VQA | 52 | 93 | 60 | 14 | 51 | 68 | 23 | 6.25 | 0.99 | **0.29** | 0.219 |
| *Binning Strategies* | | | | | | | | | | | |
| Refined Quantile Bins | 54 | 91 | 63 | 20 | 56 | 69 | 26 | **6.40** | 1.02 | **0.29** | 0.219 |
| Uniform Bins | 57 | 94 | 61 | **26** | 55 | 70 | 34 | 6.32 | 0.98 | **0.29** | 0.219 |
| *Post-Training* | | | | | | | | | | | |
| Post-Training (50k) | 56 | 93 | 62 | 23 | 56 | 67 | 32 | 6.22 | 0.88 | 0.28 | 0.217 |
| *Text Alignment* | | | | | | | | | | | |
| Only Text Alignment | 40 | 84 | 36 | 9 | 37 | 63 | 14 | 4.58 | -0.09 | 0.22 | 0.203 |

Table 2: Detailed numerical results for all ablation studies on GenEval and PartiPrompts benchmarks. **Bold** indicates best result, underline indicates second best.

## D PREPROCESSING COMPUTATIONAL COSTS

Figure 19 presents a computational breakdown of the MIRO preprocessing pipeline. Calculating rewards for the 16M image dataset constitutes 25% of the total computational budget. Captioning accounts for over 55% of the runtime. It remains the primary bottleneck. Given that reward computation requires significantly less compute than captioning, MIRO offers a efficient alternative to captioning-based approaches for low-budget training.

## E CAPTIONING DETAILS

We caption images using the model `google/gemma-3-12b-it` available on HuggingFace. To generate long captions, we use the following prompt:

```
"Analyze the following image in detail. Identify all prominent objects,
    their attributes (color, material, shape, size, texture), their
    spatial relationships, the overall scene and setting, the lighting
    conditions, and any relevant style or composition details."
```

```
"Based on your analysis, generate a caption of the image. It should be
    descriptive enough to allow a diffusion model to accurately
    reconstruct the image. Include specific details rather than general
    descriptions. For example, instead of 'a blue car,' describe it as 'a
```

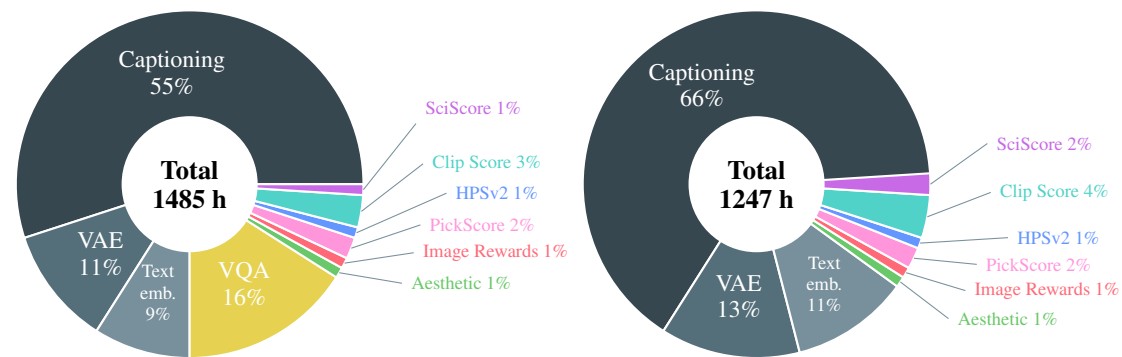

Figure 19: **The charts illustrate the time breakdown for preprocessing 16M images** (measured in H100 GPU hours). **Left**: In the full pipeline, precomputing all rewards consumes 25% of the total budget. **Right**: Removing VQA reduces the reward computation overhead to 11% (from 369 to 131 hours). In both configurations, captioning remains the primary bottleneck.

```
     shiny, dark blue vintage sedan with chrome bumpers parked on a
     cobblestone street.'

"Please ensure the caption is enclosed within <CAPTION> and </CAPTION>
    tags. "

"Example of lengths for the caption:"

"<CAPTION> A plump gray domestic shorthair cat with symmetrical white
    paws sleeps curled into a tight circle on a sunlit oak windowsill,
    its body occupying about two-thirds of the surface. The cat's head
    rests on its hind legs, with its tail wrapped neatly around its body.
     The windowsill shows distinct wood grain patterns and a sun-bleached
     patch where sunlight consistently hits. To the left, semi-sheer
    white lace curtains with a small floral pattern hang from a wooden
    rod, partially billowing inward from a 30-centimeter-wide open window
     that reveals an out-of-focus garden with green foliage. On a round
    wooden side table to the right, a transparent glass vase holds five
    pink peonies and three white snapdragons in water, with visible
    pollen grains floating on the surface. The table's surface shows
    faint circular water stains and a light dusting of pollen. Behind the
     table, an armchair with beige linen upholstery features a folded
    gray knit blanket draped over its back. A vintage wall clock with
    Roman numerals and brass hands is mounted above the windowsill.
    Sunlight streams through the window. </CAPTION> "

"<CAPTION> A Space Gray iPad Pro displays a vibrant beach sunset,
    positioned on a rustic walnut table. The attached Magic Keyboard is
    folded back, and a Apple Pencil rests diagonally across an open
    leather folio case, revealing its suede-lined interior. To the left,
    a double-walled glass mug of black coffee sits on a cork coaster with
     a thin ring of condensation and a light sprinkle of cinnamon on the
    foam. A small ceramic pot contains a jade pothos plant with six
    visible leaves, two of which trail over the table's edge. The table's
     surface shows natural wood grain variations, including a dark, heart
    -shaped knot near the center. In the background, a mid-century modern
     sofa in teal velvet has two throw pillows with geometric patterns. A
     bookshelf against the far wall holds a mix of books, a brass desk
    lamp, and a stacked stone decoration. Natural light filters through a
     casement window with slightly wavy glass panes, creating visible
    light refractions on the table. A ceiling fan casts moving shadows,
    and a seashell wind chime hangs outside the window, occasionally
    tinkling in the breeze.</CAPTION>"
```

```
"<CAPTION> A rectangular farmhouse table is covered with a pressed linen
    tablecloth (ivory with subtle gray stripes) and meticulously set for
    eight guests. Each place setting includes a plate with Wild
    Strawberry pattern, a five-piece sterling silver flatware set, an
    water goblet, and a wine glass, all arranged with precise alignment.
    A cloth napkin is folded into a rectangle and tied with a burgundy
    silk ribbon. The centerpiece is a floral arrangement in a mercury
    glass compote, featuring six red roses, four white peonies, eight
    pine sprigs, and three cinnamon sticks. Eight tapered candles in
    brass holders are placed among the flowers. Wooden dining chairs with
     navy velvet upholstery have wool throws draped over their backs. A
    wrought iron chandelier with six Edison bulbs hangs above the table,
    casting warm light that reflects off the crystal glassware. The walls
     are adorned with cedar garlands embedded with 50 white fairy lights,
     and three framed botanical prints hang in a horizontal row. In the
    background, a fireplace with a visible flame and a stack of birch
    logs adds warmth to the scene. The air smells faintly of pine,
    cinnamon, and beeswax polish.</CAPTION> "
```

```
"An alt-text corresponding to the image is: <ALT-TEXT>  </ALT-TEXT>"
```

To compute reward scores, we generate short captions of the images. We use the following prompts :

```
"Generate a short caption of the image. Please ensure the caption is
    enclosed within <CAPTION> and </CAPTION> tags. "
```

```
"Example of lengths for the caption:"
"<CAPTION>A cat sleeping on a windowsill.</CAPTION> "
"<CAPTION>A beautiful sunset over the mountains with a clear sky.</
    CAPTION> "
"<CAPTION>A group of people enjoying a picnic in the park on a sunny day
    .</CAPTION> "
"<CAPTION>A boy playing with a ball in the backyard.</CAPTION> "
```

Table 3 shows examples of long and short captions.

| Images | Captions |
|---|---|
|  | **Short:** A modern living room features a gray sofa, a vibrant watercolor garden print, and soft throws under a minimalist floor lamp.

**Long:** A bright, contemporary living room showcases a spacious gray three-seater sofa with a subtly textured, linen-blend upholstery, positioned against a pristine white wall. The wall is adorned with a large, rectangular canvas print displaying a vibrant, watercolor-inspired illustration of a whimsical garden scene, featuring a diverse array of colorful birds (including blue, orange, yellow, and green-bodied birds) amidst pink, yellow, and white flowers and lush greenery. A tall, minimalist black floor lamp with a geometric, faceted metal shade casts a soft light onto the light gray hardwood floor to the right of the sofa. A dark blue knitted throw is casually draped over the sofa's left armrest, and a lighter blue/gray throw is folded on the right. The overall aesthetic is clean, modern, and inviting, highlighting the artwork as a striking focal point in a minimalist setting. |
|  | **Short:** A scowling Santa in a velvet suit glares intensely from inside a wooden hut.

**Long:** A strikingly serious portrait of a man portraying Santa Claus, captured in a tight close-up from within a rustic wooden sauna. He is attired in a bright red Santa suit made of a textured velvet-like fabric, complete with white fur trim around the collar and cuffs, and a traditional conical hat featuring a large, plush white pom-pom. His long, thick, and unkempt white beard covers a significant portion of his face. His dark, bushy eyebrows are heavily furrowed, conveying a palpable sense of discontent or annoyance, and his dark eyes gaze directly at the viewer with alarming intensity. The sauna is constructed from light-colored pine planks, exhibiting a natural wood grain and a slightly rough texture, creating a warm but somewhat enclosed feeling. Dramatic directional lighting from the left aggressively illuminates his face, casting heavy shadows to the right, accentuating the wrinkles and emphasizing the seriousness of his expression. The overall effect is a jarring juxtaposition of the familiar Christmas icon with an unsettling and unexpected mood, suggesting a Santa Claus far removed from the joyful spirit typically associated with the holiday. |
|  | **Short:** Newlyweds share a tender embrace on a lush green lawn, she in lace and flowers, he in navy and pink.

**Long:** A heartwarming candid moment featuring a bride and groom embracing on a vibrant green lawn, set before a stately two-story white house constructed in a classic colonial architectural style with dark blue, evenly spaced shutters. The bride has light brown hair elegantly styled in an updo accented with a small white floral detail. She wears a flowing white wedding dress with a delicate lace overlay and a low, open back, revealing a glimpse of her skin. Her arms are wrapped tightly around her groom, who is dressed in a navy blue suit, a crisp white dress shirt, and a light pink tie. A shiny silver wedding band adorns his left ring finger. The bride holds a bouquet consisting of a mix of white and pale pink roses interspersed with lush greenery. The background features meticulously trimmed hedges and a mature tree with a thick, textured gray trunk. Soft, diffused natural light bathes the scene, creating gentle shadows across the lawn. The overall impression is one of joy, love, and timeless elegance, characteristic of a wedding day celebration. |

Table 3: Example of captions used in the training set.

## F  TRAINING PROGRESSION ADDITIONAL EXAMPLES

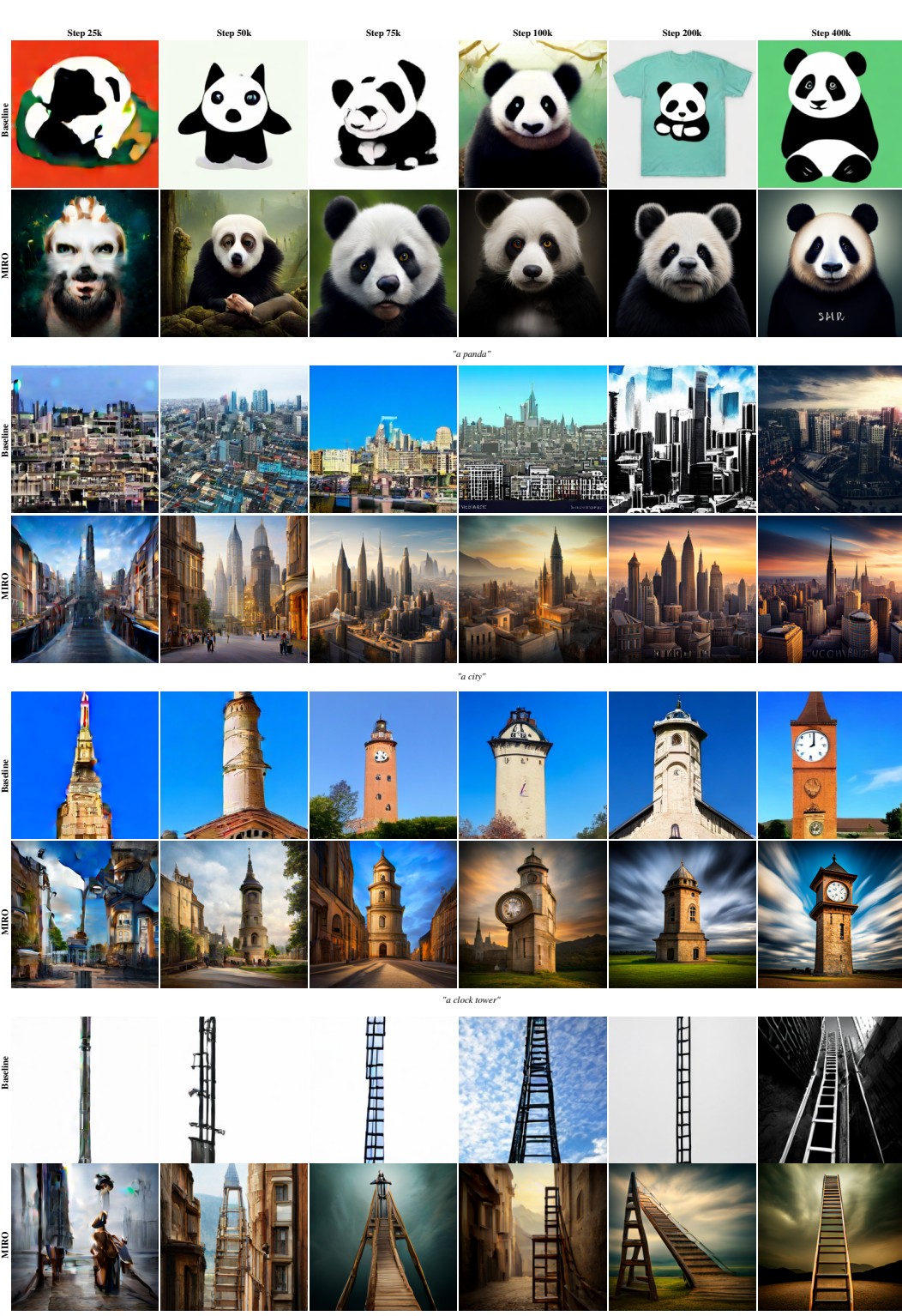

Figure 20: Additional training progression examples showing generated images at different training steps. Each row pair shows baseline (top) and MIRO (bottom) model outputs for the same prompt across training.

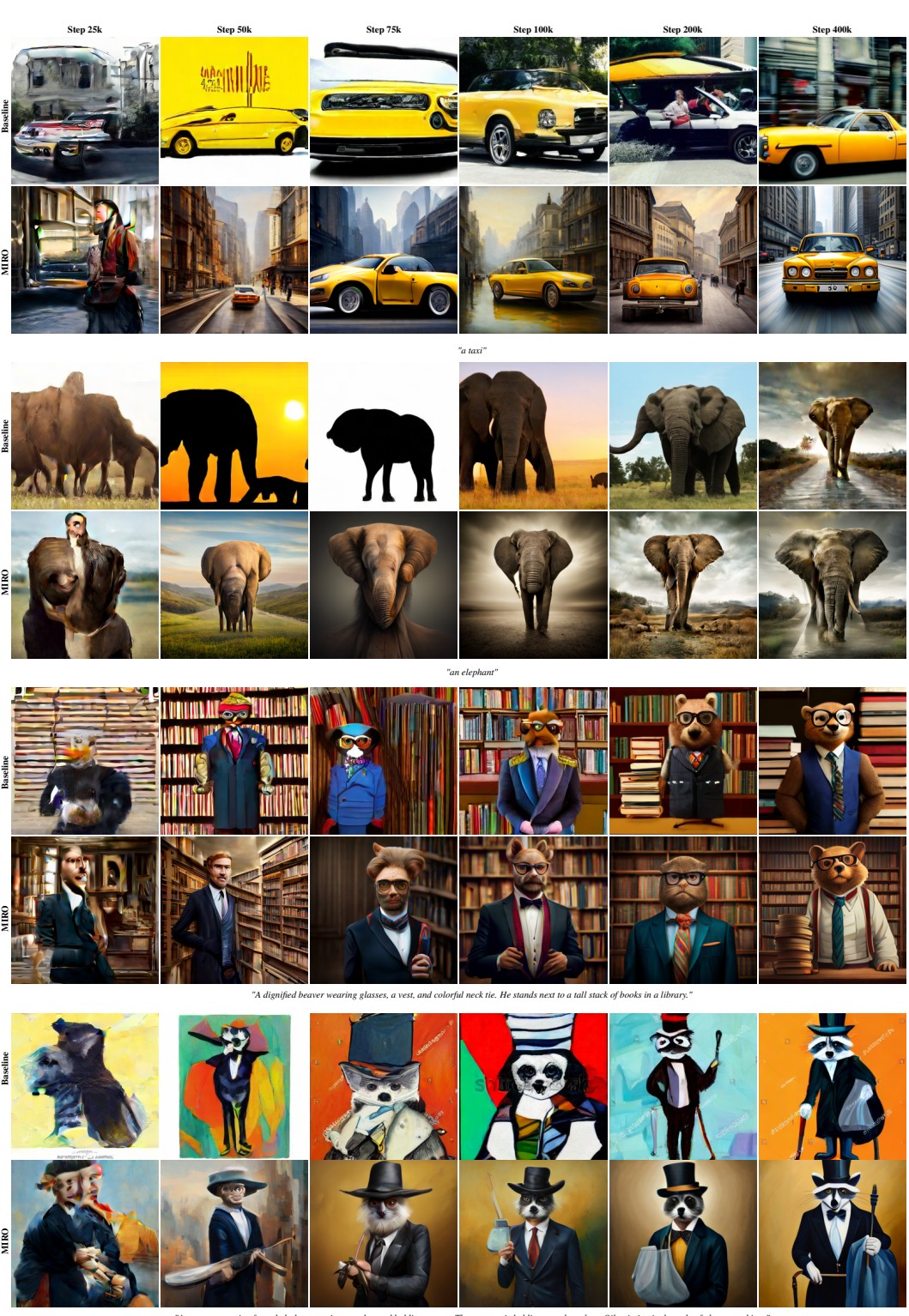

Figure 21: Additional training progression examples showing generated images at different training steps. Each row pair shows baseline (top) and MIRO (bottom) model outputs for the same prompt across training.

