# OpenReview forum: "MIRO: MultI-Reward cOnditioned pretraining improves T2I quality and efficiency"
_ICLR.cc/2026/Conference — Submitted to ICLR 2026_

### Official Review · Reviewer_DCVs · 2025-10-18

**Soundness:** 2
**Presentation:** 3
**Contribution:** 2
**Rating:** 4
**Confidence:** 5

**Summary:**

This paper introduces MIRO, a novel pretraining framework that align with user preferences directly into the text-to-image generation process, avoiding common post-hoc fine-tuning stages. The core idea is to condition the generative model (based on flow matching) on a vector of scores from multiple, diverse reward models, such as those for aesthetics, composition, and text-image correspondence. By learning from the entire quality spectrum of the data, the method significantly accelerates training convergence (up to 19x faster) and mitigates reward hacking by balancing competing objectives. Empirically, MIRO achieves state-of-the-art results on the GenEval benchmark and user preference scores, outperforming larger models while being more computationally efficient and offering explicit control over quality attributes at inference time.

**Strengths:**

paper presents an interesting and practical approach to integrating alignment into the pretraining phase of text-to-image models. The main strengths are as follows:

1. The core idea of MIRO is conceptually simple and direct. It proposes to condition the model on multiple rewards. Despite its simplicity, the method is shown to be highly effective, leading to significant improvements in both generation quality and alignment.
2. A key practical contribution is the dramatic acceleration in model convergence. The paper convincingly demonstrates (e.g., in Figure 3) that MIRO converges much faster on multiple preference scores (up to 19x faster on AestheticScore). This substantial gain in sample efficiency makes the approach computationally appealing and highlights the benefit of providing dense, multi-faceted reward signals during the initial training phase.
3. The authors have conducted a thorough evaluation across a wide array of metrics. The experiments are not limited to a few aesthetic scores but also include established user preference benchmarks (PickAScore, ImageReward, HPSv2) and a compositional reasoning benchmark (GenEval).
4. The paper provides valuable insights by exploring how MIRO interacts with other state-of-the-art techniques. The analyses on improving text-image alignment, the synergy with synthetic captions, and the benefits for test-time scaling are particularly strong.

**Weaknesses:**

While the paper presents compelling empirical results, there are several weaknesses that limit its overall contribution and impact.

1.  The primary weakness is the limited novelty of the core method. The idea of conditioning a generative model on external signals is well-established (e.g., class-conditional generation, classifier guidance). The work of Dufour et al. (2024), which the authors cite, has already explored conditioning on a single reward (CLIP score). The main contribution of MIRO is extending this concept from a single reward to a vector of multiple rewards. While the engineering and empirical results are valuable, this extension feels more like an incremental step rather than a fundamental conceptual breakthrough.

2.  The paper highlights gains in training and inference efficiency but completely omits the significant computational cost of the data preparation stage. To implement MIRO, one must run *seven* different reward models over the entire 16M image-text pair dataset. This is a massive, non-trivial upfront computational expenditure.

3.  The paper presents the multi-reward conditioning as a universally positive approach but lacks a critical discussion of its potential downsides or complexities. It is highly probable that different rewards are not always complementary and may be in direct conflict (e.g., maximizing an aesthetic score might penalize compositional correctness or text fidelity). The paper does not explore this problem.

4. The legibility of several key figures is poor, which hinders the reader's ability to fully interpret the results. Specifically, the numerical labels and legends in the radar plots of Figure 2 and Figure 5 are too small to be read comfortably without significant zooming.

5. This paper lacks REPRODUCIBILITY STATEMENT, ETHICS STATEMENT, and THE USE OF LLMS

**Questions:**

1.  The core idea builds upon prior work like Dufour et al. (2024), which conditioned on a single reward. The main contribution here appears to be the extension to a vector of multiple rewards. Could you please elaborate on the key technical or conceptual challenges that arise specifically from this multi-reward extension?

2.  The paper rightly emphasizes the impressive gains in training and inference efficiency. However, a significant computational cost is incurred upfront by annotating the 16M-sample dataset with scores from seven separate reward models. To provide a complete picture of the method's overall efficiency, could you please:
    *   Provide an estimate of this reward annotation cost (e.g., in total GPU hours)?
    *   How does this cost compare to the computational savings from faster convergence? For example, how many GPU hours were saved by the 19x faster convergence on AestheticScore?
    *   How does this annotation cost compare to the cost of a standard post-hoc alignment stage like RLHF on a similar scale?
    This information is critical for readers to perform a fair cost-benefit analysis of the MIRO framework.

3.  The current framing suggests that combining multiple rewards is always beneficial. However, it is plausible that rewards can conflict, and your own results hint at this (e.g., SciScore vs. aesthetics). I would be very interested in seeing a deeper analysis of this aspect.
    *   Could you provide a small-scale ablation study on the impact of different reward combinations? For example, showing the performance when trained with only (1) user preference rewards (HPSv2, PickScore), (2) text-alignment rewards (CLIP, VQA), or (3) a deliberately conflicting pair. This would provide invaluable insight into which rewards are most crucial and how the model handles trade-offs.
    *   What happens at inference time if a user requests conflicting high scores (e.g., maximum `AestheticScore` and maximum `SciScore`)? Could you show or describe the model's output in such a scenario?

4.  The uniform binning strategy is presented as a straightforward solution for normalization. However, reward distributions are often highly skewed, with most of the data occupying a narrow score range. Does equal-population binning lead to situations where perceptually similar scores are pushed into different bins, while very different scores (at the tails of the distribution) are grouped into the same bin? Have you experimented with other binning strategies (e.g., uniform score range binning) and, if so, how did they perform?

5.  As a final, minor suggestion, I would kindly request that you increase the font size of the axis labels, numbers, and legends in the radar plots (Figures 2 and 5) for the final version of the paper. They are currently very difficult to read and make it challenging to fully appreciate the plotted results.

---

> ### Author Response · Authors · 2025-11-27
>
> We really appreciate the reviewer's thorough insight and feedback about our work. We have addressed all the weaknesses and the questions below.
>
> ## **Addressing Weaknesses**
>
> ### **Incremental novelty**
>
> We acknowledge that MIRO builds upon conditioning mechanisms explored in Dufour et al. However, we disagree that this is merely incremental. Although both approaches rely on conditioning, they address fundamentally different challenges. CAD is designed to improve data efficiency and quality, whereas MIRO introduces a distinct set of contributions centered on controllability and training dynamics.
>
> - **Controllability:** Dufour et al. (CAD) primarily use CLIP score conditioning as a proxy for data quality. MIRO operates as a control panel. By conditioning on a decomposed vector of rewards, MIRO allows for precise, inference-time steerability. This enables users to disentangle conflicting objectives.
>
> - **Multi-Objective Alignment**: MIRO introduces a conceptual leap beyond CAD by embedding multi-objective alignment directly into pre-training. MIRO conditions on a reward vector from the very start and merges pre-training and alignment into a single stage, eliminating the need for subsequent RLHF stages.
>
>
> ### **Heavy cost of precomputation of rewards**
>
> We provide a comprehensive breakdown of the computational costs in Table A (and Figure 19 of Appendix D). Reward preprocessing is computationally negligible compared to the full training pipeline, and remains a minor component even within the preprocessing stage.
>
> Specifically, reward preprocessing accounts for only 25.0% of the total preprocessing time (369 out of 1,496 GPU hours). In contrast, Recaptioning consume the vast majority of the budget (over 55%).
>
> Despite this modest cost, MIRO yields substantially higher returns than training on recaptioned data alone (see Table 1). It improves Aesthetic Score (6.28 vs. 4.96), ImageReward (1.06 vs. 0.52), HPSv2 (0.29 vs. 0.24), and PickScore (0.220 vs. 0.211). Ultimately, when considering the total computational budget (preprocessing + training), reward calculation represents a tiny fraction of the resources, yet it drives the significant performance gains observed.
>
> **Table A: Compute cost of reward preprocessing**
>
>
> | **Stage / Component** | **Cost (GPU hours)** | **% of Preprocessing** | **% of Total Pipeline** |
> | :--- | :--- | :--- | :--- |
> | PickScore | 21.32 | 1.4% | 1.0% |
> | HPSv2 | 10.85 | 0.7% | 0.5% |
> | ImageReward | 14.49 | 1.0% | 0.7% |
> | AestheticScore | 13.60 | 0.9% | 0.6% |
> | SciScore | 19.49 | 1.3% | 0.9% |
> | CLIP | 51.74 | 3.5% | 2.5% |
> | VQA | 237.81 | 15.9% | 11.3% |
> | **Total Reward Preprocessing** | **369.30** | **24.7%** | **17.6%** |
> | | | | |
> | Captioning | 817.65 | 54.7% | 39.0% |
> | VAE Encoding | 174.22 | 11.6% | 8.3% |
> | Text Embeddings | 134.94 | 9.0% | 6.4% |
> | **Total Preprocessing** | **1,496.11** | **100.0%** | **71.4%** |
> | | | | |
> | **Training Phase** | 600.00 | - | 28.6% |
> | **TOTAL** | **2,096.11** | - | **100.0%** |
>
> ### **Conflicting rewards**
>
> We fully agree that rewards need not be complementary and can even be adversarial (aesthetics vs. text fidelity for instance). However, our empirical analyses indicate that MIRO handles such conflicts constructively during training and flexibly at inference.
>
> **Constructive Conflict During Training**: To evaluate whether conflicting rewards hinder learning, we compare:
> - (i) conditioning on a single reward (Table 1, Figure 4),
> - (ii) conditioning on all seven rewards (Table 1, Figure 4),
> - (iii) a **new ablation study conditioning on six rewards, leaving one out each time** (Appendix C.1).
>
> Table 1 shows that using all the rewards consistently outperforms all single-reward variants. The leave-one-out results (iii, see Appendix C1) further demonstrate that removing any individual reward degrades performance, even those potentially in conflict. For example, removing the Aesthetic reward not only reduces the Aesthetic score (6.28 → 5.05) but also harms unrelated metrics such as HPSv2.
>
>
> **Managing Destructive Conflicts at Inference** We also recognize that some rewards can become destructive when pushed to the extreme. For instance, aggressively maximizing the Aesthetic score can degrade other metrics: GenEval score decreases from 75 to 67 when Aesthetic score is pushed to 1 (see Figures 12–13). This is precisely where MIRO’s controllability offers a key advantage: MIRO keeps reward influences adjustable at inference time.
>
> ### **Figure 2 and Figure 5 are difficult to read**
>
> We thank the reviewer for pointing this out. We have fixed it in the pdf.
>
>
> ### **Lacks reproducibility statement, ethics statement and llm use**
>
> We thank the reviewer for pointing out about those statements. We have added them in the updated PDF.

---

> ### Author Response · Authors · 2025-11-27
>
> ## **Addressing Questions**
>
> ### **Could you please elaborate on the key technical or conceptual challenges that arise specifically from this multi-reward extension?**
>
> Please see the answer to the weakness '_Incremental novelty_'
>
> ### **Computational cost of reward preprocessing**
>
> We thank the reviewer for raising this point. Below we provide a complete breakdown of (A) the reward-annotation cost, (B) its relationship to MIRO’s training-time savings, and (C) how it compares to the cost of standard post-hoc alignment methods.
>
> **A. Reward annotation cost.**
> Please see the answer to the weakness '_Heavy cost of precomputation of rewards_'.
>
> **B. Comparison to GPU savings from faster convergence.**
>
> MIRO achieves **19× faster convergence** on _AestheticScore_. The full training phase requires **600 GPU·hours**. MIRO reaches the AestheticScore performances of the fully trained baseline (500k steps, 600 GPU hours) in only 25k steps. It corresponds to a saving of more than 550 GPU hours (91% of the training cost).
> The GPU-hours saved by faster convergence largely exceed the full reward-annotation cost.
>
> **C. Cost comparison w.r.t post-hoc methods**
> Post-hoc finetuning on a _single_ reward requires **500 A800 GPU·hours** (according to Fig. 1, of FlowDrop [1]) to reach performance comparable to MIRO (6.15 against 6.28 on Aesthetic for instance, see Table 2 of [1] and Table 1). This corresponds to **175 H100-equivalent hours** (A800 ≈ 0.7× A100, A100 ≈ 0.5× H100).
> Finetuning on **seven** rewards (the MIRO rewards) would therefore require **1,225 H100 GPU·hours**. This is roughly **4× the total cost of reward preprocessing** (369 GPU·h).
>
> ### **Conflicting rewards**
>
> Please see the answer to the weakness '_Conflicting rewards_'.
>
> ### **Binning strategies**
>
> We agree that reward distributions are highly skewed and that naïve binning strategies risk mapping perceptually similar samples to different bins or collapsing semantically different samples in the tails. To evaluate this, we conducted an ablation over several binning strategies:
>
> - **Quantile Bins** (the default MIRO method; equal-population bins).
> - **Refined Quantile Bins** (64 quantiles, with the upper 8 re-binned to reduce tail compression).
> - **Uniform Bins** (uniform bins over the raw score range, as suggested).
>
> Figure 17 compares these strategies. We find:
>
> **Refined Quantile Bins** slightly improve AestheticScore (6.40 vs. 6.28 for standard quantiles), confirming the reviewer’s intuition that additional resolution in skewed tails can help. However, this comes at the cost of lower performance on other reward metrics (notably ImageReward and PickScore). A similar trend appears in Table 2 on GenEval, where the original Quantile Bins outperform the refined version (0.57 vs. 0.54).
>
> **Uniform Range Bins** underperform the quantile-based strategies on most reward metrics. Nevertheless, their GenEval results remain comparable to standard Quantile Bins, suggesting some robustness.
>
> Across all metrics, the original **Quantile Bins** provide the most stable and well-balanced performance.  Refined tail modeling can help specific metrics but it introduces regressions elsewhere.
>
> ### **Font size**
>
> We have fixed it in the pdf. Thanks for the suggestion.
>
> [1] Liu, Jie, et al. "Flow-grpo: Training flow matching models via online rl." arXiv, 2025.

---

### Official Review · Reviewer_YhYH · 2025-10-31

**Soundness:** 2
**Presentation:** 3
**Contribution:** 2
**Rating:** 2
**Confidence:** 4

**Summary:**

This paper addresses the text-to-image pretraining problem. MIRO is proposed, which conditions model training on multiple reward models to directly learn user preferences. This method improves visual quality, accelerates training, and achieves state-of-the-art results on GenEval benchmarks and preference metrics (PickAScore, ImageReward, HPsV2).

**Strengths:**

1. The proposed approach demonstrates accelerated convergence during training, as evidenced by Figure 3, which illustrates MIRO’s significantly faster optimization compared to baseline methods.
2. MIRO consistently outperforms the baseline across all evaluated benchmarks, with Table 1 highlighting superior performance on GenEval and PartiPrompts metrics.
3. The integration of multiple reward models during pretraining represents an innovative strategy in text-to-image (T2I) generation, addressing limitations of prior single-reward optimization frameworks.

**Weaknesses:**

1. Experimental Limitations: The experimental design raises critical concerns regarding scalability and generalizability. The study focuses solely on a 0.36B parameter model—a relatively small architecture in T2I research—and trains it on only 16M image-text pairs. These constraints undermine confidence in the method’s ability to scale to industry-standard large models (e.g., 10B+ parameters) or real-world datasets. Additionally, the conclusions drawn from such limited experiments lack sufficient statistical rigor.
2. Theoretical Justification: The paper fails to provide a compelling theoretical motivation for integrating reward models into the pretraining phase. The authors do not adequately explain why this approach is inherently superior to conventional fine-tuning or reinforcement learning (RL) paradigms, leaving its necessity unproven.
3. Computational Burden: The proposed framework exhibits prohibitive resource requirements. Continuous reward annotation for all training samples becomes infeasible at scale, particularly when pretraining on billion-scale datasets. A more practical alternative would be to apply MIRO during the supervised fine-tuning (SFT) phase, where high-quality curated data could mitigate annotation costs.
4. Overstated Claims: Several assertions in the paper lack empirical validation:
* As claimed in Line 201, the statement that "MIRO eliminates the need for separate fine-tuning or RL stages" is unsubstantiated. Existing evidence demonstrates that domain-specific fine-tuning with curated datasets significantly enhances performance, while RL remains critical for optimizing text rendering. No experiments in this work refute these dependencies.
* The assertion in Line 206 that "MIRO leverages the entire quality spectrum" contradicts the authors’ own methodology, as they explicitly filter training data to CC12M and LAION Aesthetics 6+, effectively discarding low-quality samples.
5. Evaluation Scope: The empirical validation relies on an overly narrow benchmark set (GenEval and PartiPrompts). To strengthen the conclusions, the authors should evaluate MIRO on additional  benchmarks to ensure robustness across diverse modalities and use cases.

**Questions:**

Please see the weaknesses.

---

> ### Author Response · Authors · 2025-11-27
>
> We thank the reviewer for their feedback. We have addressed the weaknesses that the reviewer raised.
>
> ## **Addressing Weaknesses**
>
> ### **Undermine confidence in the method’s ability to scale to industry-standard large models and real world datasets**
>
> We show that it is possible to match the performance of a large-scale model (FLUX-dev, 12B parameters) while using only a fraction of the computational cost (training only on **16M real-world images**). This finding is significant in the long run as it establishes that **efficiency and scalability can be achieved simultaneously**.
>
> Our results highlight two key implications: (a) with MIRO, smaller models trained on substantially less data can still reach state-of-the-art performance, and (b) it promises that the same method can be applied to larger models, enabling even greater gains.
>
> Ultimately, this evidences that the "smart" use of data (rather than brute-force scaling alone) leads to competitive and highly cost-effective performance.
>
> ### **Why this approach is inherently superior to conventional fine-tuning or reinforcement learning (RL) paradigms?**
>
> We thank the reviewer for raising this important point.
>
> **Controllability vs. Distribution Shift** Conventional fine-tuning and RLHF update model weights to maximize a specific reward, It shifts the model’s entire distribution toward that bias. This results in a loss of controllability. One cannot easily "turn off" the bias or dynamically mix different rewards at inference time without re-training. In contrast, MIRO treats rewards as conditioning signals. This inherently preserves the model's versatility, allowing users to specify, mix, or dampen reward targets at inference time (See Figure 3). This capability allows a single model to adapt to various reward preferences without the need for multiple fine-tuned checkpoints.
>
> **Efficiency and "Post-Training" Performance** To further demonstrate MIRO's superiority as a distinct paradigm, we conducted a new experiment comparing MIRO pretrained from scratch against a MIRO Post-training approach (see Appendix C.3). We compare:
> - **MIRO** applied for the full 500k steps.
> - **Post-Training** A baseline model trained for 450k steps, followed by MIRO fine-tuning for only 50k steps.
>
> As illustrated in Figure 18 and Table 2, the **Post-training** approach demonstrates remarkable efficiency. Despite using MIRO for only 10% of the total training steps, it achieves parity on some Aesthetic metrics (ImageRewards, HPSv2) and on GenEval.
>
> This highlights **two distinct advantages** of MIRO compared to conventional RL stages. First, it preserves inference-time controllability: users can explicitly manage the specific rewards and biases the model prioritizes. Second, MIRO is also highly effective when applied as a fine-tuning strategy.
>
> ### **High cost for reward precomputation**
>
> We provide a comprehensive breakdown of the computational costs in Table A (and Figure 19 of Appendix D). Reward preprocessing is computationally negligible compared to the full training pipeline, and remains a minor component even within the preprocessing stage.
>
> Specifically, reward preprocessing accounts for only 25.0% of the total preprocessing time (369 out of 1,496 GPU hours). In contrast, Recaptioning consume the vast majority of the budget (over 55%).
>
> Despite this modest cost, MIRO yields substantially higher returns than training on recaptioned data alone (see Table 1). It improves Aesthetic Score (6.28 vs. 4.96), ImageReward (1.06 vs. 0.52), HPSv2 (0.29 vs. 0.24), and PickScore (0.220 vs. 0.211). Ultimately, when considering the total computational budget (preprocessing + training), reward calculation represents a tiny fraction of the resources, yet it drives the significant performance gains observed.
>
> **Table A: Compute cost of reward preprocessing**
>
>
> | **Stage / Component** | **Cost (GPU hours)** | **% of Preprocessing** | **% of Total Pipeline** |
> | :--- | :--- | :--- | :--- |
> | PickScore | 21.32 | 1.4% | 1.0% |
> | HPSv2 | 10.85 | 0.7% | 0.5% |
> | ImageReward | 14.49 | 1.0% | 0.7% |
> | AestheticScore | 13.60 | 0.9% | 0.6% |
> | SciScore | 19.49 | 1.3% | 0.9% |
> | CLIP | 51.74 | 3.5% | 2.5% |
> | VQA | 237.81 | 15.9% | 11.3% |
> | **Total Reward Preprocessing** | **369.30** | **24.7%** | **17.6%** |
> | | | | |
> | Captioning | 817.65 | 54.7% | 39.0% |
> | VAE Encoding | 174.22 | 11.6% | 8.3% |
> | Text Embeddings | 134.94 | 9.0% | 6.4% |
> | **Total Preprocessing** | **1,496.11** | **100.0%** | **71.4%** |
> | | | | |
> | **Training Phase** | 600.00 | - | 28.6% |
> | **TOTAL** | **2,096.11** | - | **100.0%** |
>
>
> ### **Overstated Claim of MIRO eliminates the need for separate fine-tuning or RL stages**
>
>
> See the answer to _"Why this approach is inherently superior to conventional fine-tuning or reinforcement learning (RL) paradigms?"_

---

> > ### Author Response · Authors · 2025-11-27
> >
> > ### **Overstated claim of MIRO leverages the entire quality spectrum**
> >
> > While it is true that LAION Aesthetics 6+ represents a filtered high-quality subset, we must clarify that we do not apply aesthetic filtering to the CC12M dataset.
> >
> > Standard training recipes typically prune "bad" data aggressively or fine-tune exclusively on high-quality subsets, limiting the model to a narrow slice of the distribution. In contrast, by retaining the unfiltered CC12M data, MIRO intentionally exposes the model to a wide variance of quality.
> >
> > The statement that "MIRO leverages the entire quality spectrum" refers to this mechanism: because the model is conditioned on the reward vector for every sample (whether high or low quality), it learns to distinguish and map the relationship between the conditioning signal and the image quality. This allows the model to understand the difference between "bad" and "good" data, rather than simply overfitting to a high-quality subset or not seeing "bad" data.
> >
> >
> > ### **Additional benchmarks: only evalutated on geneval and partiprompts**
> >
> > We thank the reviewer for this suggestio. We selected **GenEval** and **PartiPrompts** because they have become standard benchmarks in recent state-of-the-art literature (e.g. SD3-Medium, SANA or RENO [1]), and they allow direct and fair comparison against prior methods.
> >
> > However, we are eager to ensure our validation is as comprehensive as possible. Could you clarify which specific modalities or use cases you feel are missing? We would be happy to include additional standard benchmarks.
> >
> > [1] Luca Eyring, Shyamgopal Karthik, Karsten Roth, Alexey Dosovitskiy, and Zeynep Akata. Reno: Enhancing one-step text-to-image models through reward-based noise optimization. NeurIPS, 2024

---

### Official Review · Reviewer_aDYz · 2025-11-01

**Soundness:** 3
**Presentation:** 3
**Contribution:** 3
**Rating:** 6
**Confidence:** 2

**Summary:**

The paper introduces MIRO (Multi‑Reward cOnditioned Pretraining), a framework that integrates multiple reward models directly into the pretraining of text‑to‑image (T2I) models. Instead of performing post‑hoc alignment or fine‑tuning with a single reward, MIRO conditions the model on multiple reward signals spanning aesthetics, preference, semantic correspondence, and compositional reasoning. By jointly learning to map text and multiple reward targets to images, MIRO achieves controllable generation, faster convergence, and stronger generalization. Experiments on GenEval and PartiPrompts show that MIRO matches or surpasses much larger models while being 100–300× more compute‑efficient, particularly in compositional and preference alignment tasks.

**Strengths:**

* The idea of embedding multiple reward signals into pretraining is conceptually simple yet powerful, unifying data quality, efficiency, and controllability within one framework.
* Empirical results are strong and consistent: MIRO outperforms single‑reward and baseline models across aesthetic and alignment benchmarks, reaching state‑of‑the‑art GenEval and user preference scores with much lower compute.
* The method yields clear interpretability and controllability at inference time, enabling explicit adjustment of reward trade‑offs and providing an elegant alternative to complex post‑hoc RLHF pipelines.

**Weaknesses:**

* The paper lacks ablation studies analyzing sensitivity to the number and choice of reward models; it is unclear how redundant or correlated rewards affect performance or training stability.
* Although MIRO shows significant efficiency improvements, the presentation of computational cost may be incomplete. Details on hardware, batch size, and training duration are sparse, making comparisons to larger models somewhat uneven.

**Questions:**

N/A

---

> ### Author Response · Authors · 2025-11-27
>
> We appreciate that the reviewer found our work simple yet powerful and we thank the reviewer for their feedback. We have addressed the weaknesses that the reviewer raised below.
> ## **Addressing Weaknesses**
>
> ### **Ablation on choice of reward models**
>
>
> To rigorously assess the contribution of each signal, we conducted the following experiments:
> - (i) conditioning on a single reward (Table 1, Figure 4),
> - (ii) conditioning on the full vector of seven rewards (Table 1, Figure 4),
> - (iii) a **new ablation study conditioning on six rewards, leaving one out each time** (Supplementary Material C1).
>
> **Quantitative Results:** Table 1 demonstrates that multi-reward conditioning consistently outperforms single-reward baselines (MIRO improves the ImageReward improved from
> 1.04 to 1.06 and PickScore from 0.216 to 0.220).
>
> Furthermore, the leave-one-out analysis (iii) confirms that removing any single reward degrades performance across the board. For instance, excluding the Aesthetic reward reduces not only the Aesthetic score (6.28 → 5.05) but also negatively impacts HPSv2, ImageReward, and overall GenEval scores (see Table 2 and Figure 15).
>
> **Interpretation:** These findings indicate that the selected rewards cover distinct regions of the **alignment landscape**, effectively filling the "blind spots" of individual metrics. Even if a specific reward carries its own biases, its inclusion introduces a crucial constraint (e.g., regarding diversity or artifacts) that helps regularize the generation.
>
> From an optimization perspective, adding the 7th reward (or an N+1th signal) alters the **geometry of the multi-objective optimization space**. This additional constraint forces the model to explore a new region where a potential superior trade-off exists.
>
>
> ### **Presentation of computation cost is incomplete**
>
> We provide a comprehensive breakdown of the computational costs in Table A (and Figure 19 of Appendix D). Reward preprocessing is computationally negligible compared to the full training pipeline, and remains a minor component even within the preprocessing stage.
>
> Specifically, reward preprocessing accounts for only 25.0% of the total preprocessing time (369 out of 1,496 GPU hours). In contrast, Recaptioning consume the vast majority of the budget (over 55%).
>
> Despite this modest cost, MIRO yields substantially higher returns than training on recaptioned data alone (see Table 1). It improves Aesthetic Score (6.28 vs. 4.96), ImageReward (1.06 vs. 0.52), HPSv2 (0.29 vs. 0.24), and PickScore (0.220 vs. 0.211). Ultimately, when considering the total computational budget (preprocessing + training), reward calculation represents a tiny fraction of the resources, yet it drives the significant performance gains observed.
>
> **Table A: Compute cost of reward preprocessing**
>
>
> | **Stage / Component** | **Cost (GPU hours)** | **% of Preprocessing** | **% of Total Pipeline** |
> | :--- | :--- | :--- | :--- |
> | PickScore | 21.32 | 1.4% | 1.0% |
> | HPSv2 | 10.85 | 0.7% | 0.5% |
> | ImageReward | 14.49 | 1.0% | 0.7% |
> | AestheticScore | 13.60 | 0.9% | 0.6% |
> | SciScore | 19.49 | 1.3% | 0.9% |
> | CLIP | 51.74 | 3.5% | 2.5% |
> | VQA | 237.81 | 15.9% | 11.3% |
> | **Total Reward Preprocessing** | **369.30** | **24.7%** | **17.6%** |
> | | | | |
> | Captioning | 817.65 | 54.7% | 39.0% |
> | VAE Encoding | 174.22 | 11.6% | 8.3% |
> | Text Embeddings | 134.94 | 9.0% | 6.4% |
> | **Total Preprocessing** | **1,496.11** | **100.0%** | **71.4%** |
> | | | | |
> | **Training Phase** | 600.00 | - | 28.6% |
> | **TOTAL** | **2,096.11** | - | **100.0%** |

---

### Official Review · Reviewer_KiVo · 2025-11-11

**Soundness:** 3
**Presentation:** 3
**Contribution:** 3
**Rating:** 6
**Confidence:** 4

**Summary:**

This paper introduces MIRO, a framework that integrates alignment directly into the pretraining phase of text-to-image models. By conditioning the model on a vector of reward scores, MIRO learns to map desired quality levels to visual characteristics, eliminating the need for post-hoc alignment stages that can harm diversity. This method preserves the full data spectrum and enables controllable inference. Empirically, MIRO converges faster on some metrics and achieves state-of-the-art results on the GenEval benchmark, outperforming the much larger FLUX-dev model. It also demonstrates greater compute efficiency at inference than FLUX-dev.

**Strengths:**

- Converging up to 19.1x faster on AestheticScore is a massive training speedup. And the inference efficiency achieves SOTA quality with 370x less compute than FLUX-dev. It also makes Best-of-N sampling way, way cheaper.

- The paper shows (in Fig 2) that single-reward models totally overfit and tank other metrics. By training on 7 different rewards , MIRO is forced to find a healthy balance, and it ends up doing great on all of them.

- It's especially good at tough compositional tasks like Position and Color Attribution

- This method doesn't just throw away "low-quality" data, which always felt wasteful. It learns from those samples by seeing their low reward scores

**Weaknesses:**

- The whole framework is now completely dependent on the quality of your N reward models. If those models are biased or flawed, MIRO will just learn to be biased and flawed

- The paper mentions augmenting 16M images. You have to run seven different reward models over all 16M images before you can even start your faster training. That's a huge, non-trivial compute cost that has to be paid first.

**Questions:**

- How did you land on these specific 7 rewards ? Did you try a minimal set, like just one for aesthetics and one for alignment? I'm curious what the minimum viable "MIRO" looks like.

- You binned the scores using "equal population". Why that? Did you consider other ways, like having more bins for the really high-quality scores to get more fine-grained control at the top end?

---

> ### Author Response · Authors · 2025-11-27
>
> We thank the reviewer for their feedback. We have addressed the weaknesses and the questions that the reviewer raised.
>
> ## **Addressing Weaknesses**
>
> ### **Method dependent on the reward models and might include their flaws and biases**
>
> The concern that MIRO is bounded by the biases of its underlying reward models is valid. But MIRO relies on a vector of rewards specifically to mitigate the impact of individual model flaws. This stands in contrast to traditional RLHF, where the model optimizes against a single scalar reward
>
> In the single-scalar setting, if the target reward contains a flaw, the policy is incentivized to maximally exploit that specific flaw. This can lead to reward hacking [1]. In contrast, by conditioning on several rewards MIRO balances out the respective reward bias and reduces the option for reward hacking.
>
> Furthermore, MIRO offers superior controllability at inference time compared to RLHF (see Figure 3). Users can specify custom reward targets. It allows precise control over potential biases and flaws within the rewards.
>
>
> ### **Compute Cost for precomputing the rewards**
>
> We provide a comprehensive breakdown of the computational costs in Table A (and Figure 19 of Appendix D). Reward preprocessing is computationally negligible compared to the full training pipeline, and remains a minor component even within the preprocessing stage.
>
>
> **Table A: Compute cost of reward preprocessing**
>
> | **Stage / Component** | **Cost (GPU hours)** | **% of Preprocessing** | **% of Total Pipeline** |
> | :--- | :--- | :--- | :--- |
> | PickScore | 21.32 | 1.4% | 1.0% |
> | HPSv2 | 10.85 | 0.7% | 0.5% |
> | ImageReward | 14.49 | 1.0% | 0.7% |
> | AestheticScore | 13.60 | 0.9% | 0.6% |
> | SciScore | 19.49 | 1.3% | 0.9% |
> | CLIP | 51.74 | 3.5% | 2.5% |
> | VQA | 237.81 | 15.9% | 11.3% |
> | **Total Reward Preprocessing** | **369.30** | **24.7%** | **17.6%** |
> | | | | |
> | Captioning | 817.65 | 54.7% | 39.0% |
> | VAE Encoding | 174.22 | 11.6% | 8.3% |
> | Text Embeddings | 134.94 | 9.0% | 6.4% |
> | **Total Preprocessing** | **1,496.11** | **100.0%** | **71.4%** |
> | | | | |
> | **Training Phase** | 600.00 | - | 28.6% |
> | **TOTAL** | **2,096.11** | - | **100.0%** |
>
> Specifically, reward preprocessing accounts for only 25.0% of the total preprocessing time (369 out of 1,496 GPU hours). In contrast, Recaptioning consume the vast majority of the budget (over 55%).
>
> Despite this modest cost, MIRO yields substantially higher returns than training on recaptioned data alone (see Table 1). It improves Aesthetic Score (6.28 vs. 4.96), ImageReward (1.06 vs. 0.52), HPSv2 (0.29 vs. 0.24), and PickScore (0.220 vs. 0.211). Ultimately, when considering the total computational budget (preprocessing + training), reward calculation represents a tiny fraction of the resources, yet it drives the significant performance gains observed.

---

> > ### Author Response · Authors · 2025-12-03
> >
> > ## 2. Reviewer-Specific Resolutions
> >
> > ### Reviewer KiVo
> > * **Feedback:** Praised speedup (19x) and inference efficiency. Concerned about reward model bias and pre-computation cost.
> > * **Resolution:**
> >     * **Bias:** We argued that conditioning on a *vector* of rewards prevents overfitting to a single reward's flaws (reward hacking), unlike scalar RLHF.
> >     * **Cost:** Addressed via **Table A** (Preprocessing is negligible vs. recaptioning).
> >
> > ### Reviewer aDYz
> > * **Feedback:** Strong empirical results. Requested ablations on reward choices and more details on compute.
> > * **Resolution:**
> >     * **Ablations:** We added the "leave-one-out" study (Appendix C.1, **Table 2**, **Figure 15**) confirming all 7 rewards are necessary.
> >     * **Compute:** Provided **Table A** breakdown.
> >
> > ### Reviewer YhYH
> > * **Feedback:** Concerns about scalability (small model size), lack of theory, and claims that MIRO "eliminates" RLHF.
> > * **Resolution:**
> >     * **Scalability:** We matched FLUX-dev (12B params) performance using only 16M images, proving efficiency allows smaller models to compete with larger ones.
> >     * **Vs. RLHF:** We added a "Post-Training" experiment (Appendix C.3) showing MIRO is also effective as a fine-tuning strategy (50k steps), achieving parity on Aesthetic metrics.
> >     * **"Entire Spectrum":** Clarified that we use unfiltered CC12M data, enabling the model to learn the difference between "good" and "bad" via conditioning, rather than filtering data out.
> >
> > ### Reviewer DCVs
> > * **Feedback:** Questioned novelty (vs. CAD), compute cost, and conflicting rewards.
> > * **Resolution:**
> >     * **Novelty:** We clarified that while CAD uses single-reward conditioning for data efficiency, MIRO uses multi-reward vectors for *controllability* and alignment, merging pre-training and alignment into one stage.
> >     * **Conflicts:** Addressed via the "leave-one-out" ablation; removing even conflicting rewards degrades overall performance.
> >     * **Legibility:** Fixed **Figure 2** and **Figure 5** font sizes in the PDF.

---

> ### Author Response · Authors · 2025-11-27
>
> ## **Addressing Questions**
>
> ### **Why these seven specific rewards and whats the minimum viable Miro looks like**
>
> The selection of these seven rewards is grounded in recent literature [2], as they provide popular and complementary aesthetic measures. Determining the minimal optimal set of rewards is computationally intractable, as it would require an exhaustive search over the space of reward subsets. To understand the affect of each reward and the combinations we conducted the following experiments:
> - (i) conditioning on a single reward (Table 1, Figure 4),
> - (ii) conditioning on all seven rewards (Table 1, Figure 4),
> - (iii) a **new ablation study conditioning on six rewards, leaving one out each time** (Appendix C.1).
>
> As expected, the "leave-one-out" experiments (iii) resulted in lower scores on the specific metric removed (see Table 2 and Figure 15) and typically led to lower overall GenEval scores (see Table 2 and Figure 16).
>
> These results indicate that rewards can be complementary. The model effectively exploits their distinct properties to cover a broader range of the alignment landscape. This suggests our method is inherently scalable: the inclusion of future rewards capturing novel properties (e.g., particle dynamics) would likely further improve performance by enriching the conditioning signal.
>
>
> ### **Why bin with equal proportion and not more on the high end spectrum**
>
> Thank you!
> We followed the reviewer’s suggestion and ablated on different binning strategies:
> - **Quantile Bins** (as the original MIRO method)
> - **Refined Quantile Bins** (64 quantiles with the last 8 re-binned)
> - **Uniform Bins**
>
> Figure 17 compares the results of the three approaches. It shows that **Refined Quantile Bins** improves the Aesthetic Score compared to the **Quantile Bins** (6.40 vs. 6.28) but decreases the other rewards (especially ImageReward and PickScore). Similarly Table 2 shows the same trend on GenEval: the original MIRO **Quantile Bins** strategy performs better than the **Refined Quantile Bins** (0.57 vs. 0.54).
>
> Although the **Uniform Bins** strategy performs worse than quantile-based strategies on reward metrics, it achieves GenEval scores comparable to the **Quantile Bins** strategy.
>
> Considering the trade-offs across all metrics, the original **Quantile Bins** strategy remains the most robust and best-performing approach.
>
>
> [1] Liu, Jie, et al. "Flow-grpo: Training flow matching models via online rl." arXiv, 2025.
>
> [2] Eyring, Luca, et al. "Reno: Enhancing one-step text-to-image models through reward-based noise optimization." NeurIPS, 2024.

---

### Author Response · Authors · 2025-12-03
**Rebuttal Summary**

Dear new AC,

To try to help you with this change of review process, we drafted this rebuttal summary, we hope it help you in your decision process.

## Overview of Revisions & New Experiments
In response to reviewer feedback, we have updated the manuscript and conducted additional experiments to address concerns regarding computational costs, reward conflicts, and binning strategies.

**Key Updates to the PDF:**
* **New Ablation Studies:**
    * **"Leave-one-out" analysis:** Demonstrating the necessity and complementarity of the 7 rewards (Appendix C.1).
    * **Binning Strategies:** Comparing Quantile Bins vs. Uniform/Refined Bins.
    * **Post-Training:** Comparing MIRO pre-training vs. MIRO fine-tuning (Appendix C.3).
* **Formatting & Ethics:** Improved legibility of **Figure 2** and **Figure 5** (radar plots) and included Ethics/Reproducibility statements.

---

## 1. Response to Common Concerns

### A. Computational Cost of Reward Preprocessing
**Concern:** Reviewers **KiVo, aDYz, YhYH, and DCVs** queried the cost of annotating the training set (16M images) with 7 reward models, suggesting it might be prohibitive.

**Response:**
We provided a detailed breakdown demonstrating that reward preprocessing is negligible compared to the total pipeline (specifically recaptioning) and is outweighed by training efficiency gains.

* **Cost Breakdown:** Reward annotation accounts for only **25.0%** of preprocessing and **17.6%** of the total pipeline. In contrast, Recaptioning consumes **54.7%** of the preprocessing budget.
* **Net Savings:** MIRO converges **19x faster** on AestheticScore. This saves ~550 GPU hours in training, which exceeds the 369 GPU hours required for reward annotation.
* **Comparison to RLHF:** Fine-tuning on 7 rewards using standard post-hoc methods would cost ~1,225 H100 GPU-hours (approx. 4x the cost of MIRO's reward preprocessing).

**Table A: Compute Cost Analysis**

| Stage / Component | Cost (GPU hours) | % of Preprocessing | % of Total Pipeline |
| :--- | :--- | :--- | :--- |
| **Total Reward Preprocessing** | **369.30** | **24.7%** | **17.6%** |
| *(Includes PickScore, HPSv2, VQA, etc.)* | | | |
| Captioning | 817.65 | 54.7% | 39.0% |
| VAE Encoding | 174.22 | 11.6% | 8.3% |
| Text Embeddings | 134.94 | 9.0% | 6.4% |
| **Total Preprocessing** | **1,496.11** | **100.0%** | **71.4%** |
| **Training Phase** | **600.00** | - | **28.6%** |
| **TOTAL** | **2,096.11** | - | **100.0%** |

### B. Reward Conflicts and Selection
**Concern:** Reviewers **KiVo and DCVs** asked if maximizing one reward (e.g., Aesthetics) hurts others (e.g., Fidelity), and why these specific 7 rewards were chosen.

**Response:**
* **Complementarity:** A "leave-one-out" ablation (Appendix C.1) showed that removing *any* single reward degrades performance across the board. For example, removing the Aesthetic reward reduced the HPSv2 score, indicating that the rewards cover "blind spots" of individual metrics.
* **Inference Control:** While rewards can conflict if pushed to extremes (e.g., max Aesthetics hurting GenEval), MIRO allows users to dynamically adjust these weights at inference time to manage trade-offs, unlike fixed RLHF models.

### C. Binning Strategy
**Concern:** Reviewers **KiVo and DCVs** questioned the "Equal Population" (Quantile) binning strategy given the skewed distribution of reward scores.

**Response:**
We ablated three strategies:
1.  **Quantile Bins (Default):** Best overall balance.
2.  **Refined Quantile:** Slightly better AestheticScore (6.40 vs 6.28) but degraded ImageReward and PickScore.
3.  **Uniform Bins:** Underperformed on reward metrics.
*Conclusion:* The original Quantile Bins strategy remains the most robust.

---

---

### Meta-Review · Area_Chair_zeq3 · 2026-01-07

**Summary:**

The original review scores are 2,4,6,6. And the major concerns raised by reviewers are:

--If those reward models are biased or flawed, MIRO will just learn to be biased and flawed

--Computation cost of running inference multiple rewards

--Lacks ablation studies analyzing sensitivity to the number and choice of reward models

--Experiment setting like only using 0.36B parameter model with 16M training pairs

--No theoretical justification

--Several assertions in the paper lack empirical validation

--The empirical validation relies on an overly narrow benchmark set (GenEval and PartiPrompts)

--Limited novelty of the core method compared to Dufour et al. (2024)

--Downsides of using multiple rewards

**Reviewer Concerns:**

Concerns that are addressed by the rebuttal:

--Computation cost of running inference multiple rewards

--Downsides of using multiple rewards

--Lacks ablation studies analyzing sensitivity to the number and choice of reward models



Concerns that are not well addressed by the rebuttal:

-Biased or flawed reward model: authors argues that multiple rewards can mitigate the issue of flawed rewards, and better than single reward post-training. But the proposed method should be compared to multi-reward post-training instead of single reward post-training. Moreover, authors claim users can control the trade-off of these flawed multiple rewards, but how to conduct the tradeoff in practice (e.g., what are optimal tradeoff for each sample) is unclear.

--Limited novelty of the core method compared to Dufour et al. (2024): the novelty of extending from single reward to multiple reward might not be very high

--No theoretical justification

--Overly narrow benchmark set (GenEval and PartiPrompts)

**Reviewer Scores:**

The score 2 and 4 might be increased, but I don't the scores can be increased high enough to accept the paper.

---

### Decision · Program_Chairs · 2026-01-26

Reject